# Probabilistic Learning to Defer: Handling Missing Expert's Annotations and Controlling Workload Distribution

**Cuong Nguyen** [ID]
Centre for Vision, Speech and Signal Processing
University of Surrey, United Kingdom

**Thanh-Toan Do** [ID]
Department of Data Science and AI
Monash University, Australia

**Gustavo Carneiro** [ID]
Centre for Vision, Speech and Signal Processing
University of Surrey, United Kingdom

## Abstract

Recent progress in machine learning research is gradually shifting its focus towards *human-AI cooperation* due to the advantages of exploiting the reliability of human experts and the efficiency of AI models. One of the promising approaches in human-AI cooperation is *learning to defer* (L2D), where the system analyses the input data and decides to make its own decision or defer to human experts. Although L2D has demonstrated state-of-the-art performance, in its standard setting, L2D entails a severe limitation: all human experts must annotate the whole training dataset of interest, resulting in a time-consuming and expensive annotation process that can subsequently influence the size and diversity of the training set. Moreover, the current L2D does not have a principled way to control workload distribution among human experts and the AI classifier, which is critical to optimise resource allocation. We, therefore, propose a new probabilistic modelling approach inspired by the mixture-of-experts, where the Expectation - Maximisation algorithm is leverage to address the issue of missing expert's annotations. Furthermore, we introduce a constraint, which can be solved efficiently during the E-step, to control the workload distribution among human experts and the AI classifier. Empirical evaluation on synthetic and real-world datasets shows that our proposed probabilistic approach performs competitively, or surpasses previously proposed methods assessed on the same benchmarks.

## 1 Introduction

Human experts have the remarkable ability to learn new concepts and make accurate predictions. However, in many applications where the volume of data scales up exponentially, like radiology (The Royal College of Radiologists, 2023), relying solely on human experts becomes economically impractical or even infeasible due to significant increases in hiring costs and processing time. In contrast, machine learning or AI models excel at processing large amounts of information but may be prone to biases. Thus, an ideal system would therefore consist of neither human experts nor AI models in isolation but rather a judicious combination of both, leveraging the reliability of humans and the efficiency of AI models.

Such an ideal system must be capable of identifying the strengths and weaknesses of each human expert and AI model, so that samples can be assigned to decision-makers with high predictive accuracy at low processing cost. *Learning to defer* (L2D) (Madras et al., 2018) is one of the learning paradigms that provide such a capability. L2D extends *learning to reject* (Chow, 1957; Cortes et al., 2016) by deferring samples to human experts when the AI model is not confident. Although L2D can surpass the performance of human experts or the AI model alone, the learning paradigm has a notable limitation: **every human expert must annotate every sample within the training dataset**. This requirement places a substantial burden on the human experts involved, resulting in a resource-

intensive and time-consuming annotation process. Consequently, this may reduce the size of the training set, in addition to decreasing the diversity of labelling and potentially increasing biases due to the probable involvement of fewer annotators.

Another problem in L2D is the lack of workload control, where some experts (either human or AI model) are more frequently queried to make decisions, while the remaining experts are entirely ignored. Although solely relying on a few human experts is desirable in some applications where rewards are proportional to the amount of work (e.g., crowd-sourcing), it becomes unfair in fields like radiology, where payment is fixed. In such cases, the more accurate the radiologist is, the more work they are burdened with without corresponding compensation. Moreover, in those high-stakes applications, overloading one expert can easily lead to fatigue, increasing the risk of errors, and resulting in potentially poor life-and-death decisions. One typical example is reported in (Berlin, 2000), where a radiologist misdiagnosed because he interpreted 162 cases in one day while the average interpretation was only 50 cases. One workaround solution for the imbalanced workload distribution is to integrate a specific cost for each expert (e.g., proportional to salary) to balance out the workload (Narasimhan et al., 2022; Zhang et al., 2024). Such a solution is impractical because human experts will most likely be paid similar salaries, making the quantification of cost per expert infeasible. Furthermore, our preliminary experiments have shown that such a naive solution may need many iterations to fine-tune the cost to converge to the desirable workload distribution or may not converge at all in several cases. Furthermore, as L2D trades off the reliability of human experts for the efficiency of AI classifiers, it is crucial to have the ability to control the workload distribution between AI and human experts to study that trade-off. Despite such importance, controlling workload is, however, overlooked in the L2D literature. The current practice is to employ a post-hoc method by sorting the deferral score of a trained L2D method evaluated on the test set and varying the threshold to obtain different deferral rates for the AI model (Mozannar et al., 2023). Such a post-hoc approach is impractical because it requires access to all test samples before making a prediction, and it does not allow for workload control to be set during training and tested during evaluation. Hence, this prompts further studies to develop a principled mechanism to control the workload distribution in L2D.

We, therefore, propose a probabilistic framework inspired by the mixture-of-experts with constraints to address both problems: handling missing expert's annotation and controlling workload distribution. Treating the missing expert's annotation as a latent variable enables us to formulate the proposed framework with the Expectation - Maximisation (EM) algorithm. In addition, the constraints introduced to control the workload distribution among experts and between AI and human experts yield an efficient solution, and hence, can be integrated seamlessly into the EM algorithm without compromising computational efficiency. Our contributions can be summarised as follows:

- We propose a new probabilistic modelling approach for L2D with missing expert's annotations and leverage the EM algorithm to optimise it. Such a solution extends the capacity of L2D to practical settings where each expert annotates only a portion of the training dataset.

- We also introduce a constraint in the optimisation of the E step, which can be solved efficiently, to control the workload distributed to each human expert and the AI model.

The empirical evaluation on both synthetic and real-world datasets at different missing rates demonstrates the capability of our proposed method when dealing with missing expert's annotation settings, while being able to control the desired workload for human experts and the AI model.

## 2 BACKGROUND

Learning to defer (Madras et al., 2018) is motivated from the observation that in certain scenarios or datasets, **a machine learning model trained on ground truth cannot perform on-par with human experts**. Hence, L2D is proposed by integrating a deferral or gating mechanism to query either a human expert or an AI classifier to make decisions. In other words, L2D is similar to a mixture of $M$ human experts and an AI classifier. The AI classifier is

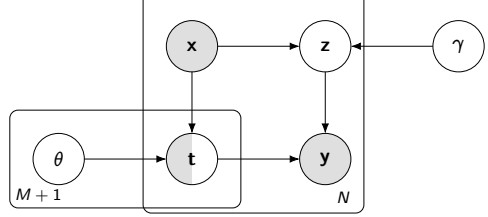

Figure 1: L2D is modelled as a mixture of $M$ human experts and an AI classifier. The human expert's annotation $\mathbf{t}$ is either observed (standard L2D) or hidden (our setting).

learnt from ground truth data to support the classification for the easy examples, reducing the cost of querying human experts.

A $C$-way classification task in LDA consists of a dataset $\mathcal{D} = (\{(\mathbf{x}_i, \mathbf{y}_i, (\mathbf{t}_i^{(m)})_{m=1}^M)\}_{i=1}^N$, where $\mathbf{x}_i \in \mathcal{X} \subseteq \mathbb{R}^d$ is an input sample, $\mathbf{y}_i \in \mathcal{Y} = \{1, \ldots, C\}$ is the corresponding ground truth label, and $\mathbf{t}_i^{(m)} \in \mathcal{Y}$ represents the annotation of sample $\mathbf{x}_i$ made by an expert indexed by $m$. In addition, an AI classifier, parameterised by $\theta_{M+1}$, is introduced as a learnable expert.

We follow the mixture of experts to model the L2D problem. The training data of L2D can be modelled as follows:

1. draw a sample from data distribution: $\mathbf{x} \sim \Pr(\mathbf{x})$,

2. draw an annotation from each expert: $\mathbf{t}^{(m)} \sim \Pr(\mathbf{t}|\mathbf{x}, \theta_m) = \text{Categorical}(\mathbf{t}|f(\mathbf{x}; \theta_m))$,

3. draw a categorical variable to select an expert: $\mathbf{z} \sim \Pr(\mathbf{z}|\mathbf{x}, \gamma) = \text{Categorical}(\mathbf{z}|g(\mathbf{x}; \gamma))$,

4. draw a label from the belief of that expert: $\mathbf{y} \sim \Pr(\mathbf{y}|\mathbf{z}, \mathbf{t}) = \text{Categorical}(\mathbf{y}|\mathbf{t}^{(z)})$,

where: $g(.; \gamma) : \mathcal{X} \to \{1, \ldots, M + 1\}$ is the gating model, and $f(.; \theta_m) : \mathcal{X} \to \Delta_{C-1} = \{\mathbf{v} \in [0, 1]^C : \mathbf{v}^\top \mathbf{1} = 1\}$ is a classification model representing an expert (including the AI classifier) indexed by $m \in \{1, \ldots, M + 1\}$. Note that in this setting, **every sample in the training set is annotated by all experts**, where the function representing each human expert $f(.; \theta_m)$ is simply a mapping table that returns an annotation $\mathbf{t}_i^{(m)}$ given an input $\mathbf{x}_i$, with $\mathbf{t}_i^{(m)} = f(\mathbf{x}_i; \theta_m), \forall m \in \{1, \ldots, M\}$. The data modelling can also be illustrated through the graphical model in Fig. 1.

The objective of L2D is to learn the gating model's and AI classifier's parameters by maximising data-likelihood on observed data, which can be written as follows:

$$\max_{\gamma, \theta_{M+1}} \sum_{i=1}^N \ln \Pr\left(\mathbf{y}_i, \prod_{m=1}^{M+1} \mathbf{t}_i^{(m)} \middle| \mathbf{x}_i, \gamma, \{\theta\}_{m=1}^{M+1}\right)$$
$$= \max_{\gamma, \theta_{M+1}} \sum_{i=1}^N \ln \Pr\left(\mathbf{y}_i \middle| \mathbf{x}_i, \prod_{m=1}^{M+1} \mathbf{t}_i^{(m)}, \gamma\right) + \ln \Pr\left(\mathbf{t}_i^{(M+1)} \middle| \mathbf{x}_i, \theta_{M+1}\right). \quad (1)$$

The first term on the right hand side of Eq. (1) is to train the gating model, parameterised by $\gamma$, while the second term is used to train the AI classifier. The annotation $\mathbf{t}$ of the AI classifier, therefore, has to be modified accordingly: in the first term, $\mathbf{t}$ is the prediction of the AI classifier: $\mathbf{t}_i^{(M+1)} = f(\mathbf{x}_i; \theta_{M+1})$, so that the gating model can assess the performance of each expert (including human and the AI classifier) to defer, while in the second term, it is the ground truth label: $\mathbf{t}_i^{(M+1)} = \mathbf{y}_i$ in order to train the AI classifier. Such a modification is required because L2D has both un-learnable (i.e., human) and learnable experts (i.e., the classifier). Equivalently, one can remove the second term from Eq. (1) by training the classifier on ground truth labels separately, then considering that trained classifier as an un-learnable human expert. In that case, Eq. (1) is simplified to consist of only the first term, where the annotation $\mathbf{t}_i^{(M+1)}$ does not need to be modified.

In this modelling, L2D is a latent variable model, where the latent variable is the index $\mathbf{z}$ of the expert being selected. Hence, the objective in Eq. (1) can be optimised by the EM algorithm as shown in Appendix A.1.

## 3 MISSING EXPERT'S ANNOTATIONS

In this section, we relax the assumption that every sample in the training set is annotated by all human experts, presented in Section 2. In particular, for each training sample, annotations made by some human experts are assumed to be observed, while the ones made by the remaining experts are unobserved (or missing).

For each sample, we denote $\mathcal{D}_i^{\text{obs.}}$ as the set of expert indices that annotate sample $\mathbf{x}_i$, and $\mathcal{D}_i^{\text{unobs.}}$ as the set of expert indices who do not annotate sample $\mathbf{x}_i$. The objective function originally presented in Eq. (1) that maximises log-likelihood on observed data can be rewritten as follows:

$$\max_{\gamma, \{\theta_m\}_{m=1}^{M+1}} \sum_{i=1}^N \ln \Pr\left(\mathbf{y}_i, \prod_{m \in \mathcal{D}_i^{\text{obs.}}} \mathbf{t}_i^{(m)} \middle| \mathbf{x}_i, \gamma, \{\theta\}_{m=1}^{M+1}\right). \quad (2)$$

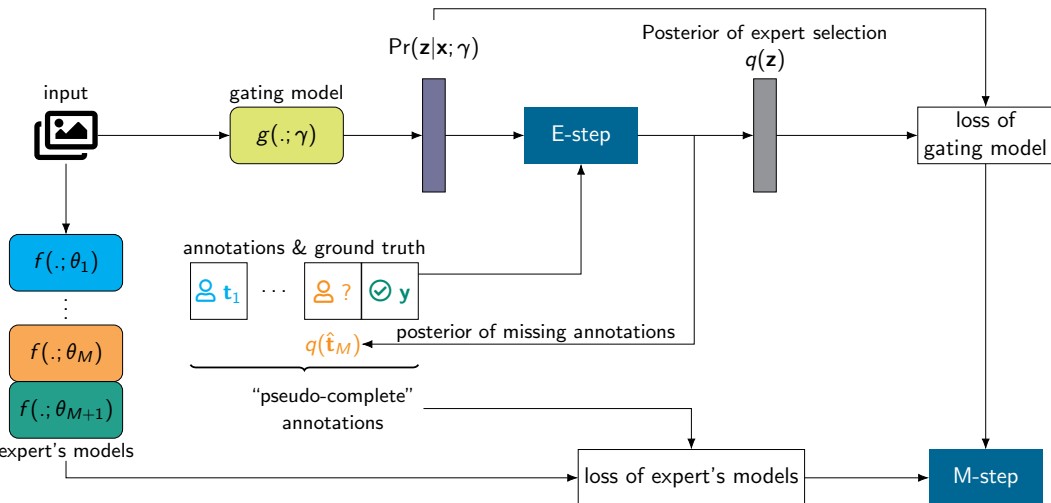

Figure 2: Visualisation of the proposed probabilistic L2D where the E-step infers the posteriors of expert selection and missing annotations, while the M-step maximises the "completed" data likelihood to estimate the gating function and AI model parameters $\gamma$ and $\theta_{M+1}$.

The objective function in Eq. (2) learns both the gating model parameter $\gamma$ and the parameter of $M$ experts plus the AI classifier, denoted by $\{\theta_m\}_{m=1}^{M+1}$, while the standard (or "complete-annotation") L2D in Eq. (1) learns only the parameters of the gating model and classifier.

Due to the latent variables $\mathbf{z}$ and $\mathbf{t}^{(j)}$, for $j \in \mathcal{D}^{\text{unobs.}}$, it is difficult to calculate the log-likelihood in Eq. (2). Thus, the EM algorithm is employed to optimise a lower-bound of the objective function in Eq. (2). The "complete" data log-likelihood in Eq. (2) for one sample can be written as follows:

$$
\begin{aligned}
Q_i = {} & \mathbb{E}_{q(\mathbf{z}_i, \prod_{j \in \mathcal{D}_i^{\text{unobs.}}} \mathbf{t}_i^{(j)})} \ln \Pr\Big( \mathbf{y}_i, \textstyle\prod_{m \in \mathcal{D}_i^{\text{obs.}}} \mathbf{t}_i^{(m)}, \prod_{j \in \mathcal{D}_i^{\text{unobs.}}} \mathbf{t}_i^{(j)}, \mathbf{z}_i \Big| \mathbf{x}_i, \gamma, \{\theta_m\}_{m=1}^{M+1} \Big) \\
= {} & \mathbb{E}_{q(\mathbf{z}_i, \prod_{j \in \mathcal{D}_i^{\text{unobs.}}} \mathbf{t}_i^{(j)})} \Big[ \ln \Pr\Big( \mathbf{y}_i \Big| \mathbf{z}_i, \textstyle\prod_{m \in \mathcal{D}_i^{\text{obs.}}} \mathbf{t}_i^{(m)}, \prod_{j \in \mathcal{D}_i^{\text{unobs.}}} \mathbf{t}_i^{(j)} \Big) \\
& + \textstyle\sum_{j \in \mathcal{D}_i^{\text{unobs.}}} \ln \Pr\Big( \mathbf{t}_i^{(j)} \Big| \mathbf{x}_i, \theta_j \Big) + \ln \Pr(\mathbf{z}_i | \mathbf{x}_i, \gamma) \Big] + \sum_{m \in \mathcal{D}_i^{\text{obs.}}} \ln \Pr\Big( \mathbf{t}_i^{(m)} \Big| \mathbf{x}_i, \theta_m \Big), \quad (3)
\end{aligned}
$$

where $q(\mathbf{z}_i, \prod_{j \in \mathcal{D}_i^{\text{unobs.}}} \mathbf{t}_i^{(j)})$ denotes the posterior of the latent variables.

**E step** The posterior of both $\mathbf{z}$ and $\mathbf{t}^{(j)}$ can be obtained through a mean-field variational inference as follows:

$$
\begin{aligned}
q\Big( \mathbf{z}_i, \textstyle\prod_{j \in \mathcal{D}_i^{\text{unobs.}}} \mathbf{t}_i^{(j)} \Big) & = q(\mathbf{z}_i) \textstyle\prod_{j \in \mathcal{D}_i^{\text{unobs.}}} q\Big( \mathbf{t}_i^{(j)} \Big) = \text{argmin}_q \, \text{KL} \Big[ q(\mathbf{z}_i) \textstyle\prod_{j \in \mathcal{D}_i^{\text{unobs.}}} q\Big( \mathbf{t}_i^{(j)} \Big) \\
& \Big\| \Pr\Big( \mathbf{z}_i, \textstyle\prod_{j \in \mathcal{D}_i^{\text{unobs.}}} \mathbf{t}_i^{(j)} \Big| \mathbf{x}_i, \mathbf{y}_i, \prod_{m \in \mathcal{D}_i^{\text{obs.}}} \mathbf{t}_i^{(m)}, \gamma^{(k)}, \{\theta_m^{(k)}\}_{m=1}^{M+1} \Big) \Big] \\
= {} & \underset{q}{\text{argmin}} \, \text{KL} \Big[ q(\mathbf{z}_i) \| \Pr\Big( \mathbf{z}_i | \mathbf{x}_i, \gamma^{(k)} \Big) \Big] + \sum_{j \in \mathcal{D}_i^{\text{unobs.}}} \text{KL} \Big[ q(\mathbf{t}_i^{(j)}) \Big\| \Pr\Big( \mathbf{t}_i^{(j)} | \mathbf{x}_i, \theta_j^{(k)} \Big) \Big] \\
& - \mathbb{E}_{q(\mathbf{z}_i) \prod_{j \in \mathcal{D}_i^{\text{unobs.}}} q\left( \mathbf{t}_i^{(j)} \right)} \Big[ \ln \Pr\Big( \mathbf{y}_i \Big| \mathbf{z}_i, \textstyle\prod_{m \in \mathcal{D}_i^{\text{obs.}}} \mathbf{t}_i^{(m)}, \prod_{j \in \mathcal{D}_i^{\text{unobs.}}} \mathbf{t}_i^{(j)} \Big) \Big],
\end{aligned}
$$

$$(4)$$

where $\gamma^{(k)}$ and $\theta^{(k)}$ denote the gating and expert model parameters at the $k$-th iteration.

The optimisation in Eq. (4) can be expanded further and solved by the fixed-point iteration method. Please refer to Appendix B for the detailed derivation of the solution. Note that the log-likelihood of the last term in Eq. (4) may lead to numerical instability when $\mathbf{t}$ is a hard label (i.e., when $\mathbf{t}$ is observed). In our implementation, we smooth the observed annotations when calculating that log-likelihood to avoid computational instability issue.

**M step**  The posteriors of latent variables obtained in the E step allows the calculation of the complete data log-likelihood in Eq. (3). In the M step, that completed data log-likelihood is maximised with respect to the parameters of interest $\gamma$ and $\theta$ as follows:

$$
\begin{aligned}
\gamma^{(k+1)}, \{\theta_m^{(k+1)}\}_{m=1}^{M+1} &\leftarrow \operatorname{argmax}_{\gamma, \{\theta_m\}_{m=1}^{M+1}} \sum_{i=1}^{N} Q_i(\gamma, \theta, \gamma^{(k)}, \{\theta_m^{(k)}\}_{m=1}^{M+1}) \\
&= \operatorname{argmax}_{\gamma, \{\theta_m\}_{m=1}^{M+1}} \sum_{i=1}^{N} \sum_{j \in \mathcal{D}_i^{\text{unobs.}}} \mathbb{E}_{q(\mathbf{t}_i^{(j)})} \left[ \ln \Pr\left( \mathbf{t}_i^{(j)} \Big| \mathbf{x}_i, \theta_j \right) \right] \\
&\quad + \sum_{m \in \mathcal{D}_i^{\text{obs.}}} \ln \Pr\left( \mathbf{t}_i^{(m)} \Big| \mathbf{x}_i, \theta_m \right) + \mathbb{E}_{q(\mathbf{z}_i)} \left[ \ln \Pr(\mathbf{z}_i | \mathbf{x}_i, \gamma) \right].
\end{aligned}
\tag{5}
$$

Note that the term related to $\Pr(\mathbf{y}|\mathbf{z}, \mathbf{t})$ in Eq. (3) is omitted in the M step because it is considered as a constant w.r.t. $\gamma$ and $\{\theta_m\}_{m=1}^{M+1}$. The whole training procedure can be visualised in Fig. 2.

**Remark 1** *The first and the second terms in Eq.* (5) *resemble the minimum entropy principle in semi-supervised learning for expert models parameterised by* $\{\theta_m\}_{m=1}^{M}$ (*Grandvalet & Bengio, 2004*). *In particular, the first term is equivalent to minimising the entropy of the missing expert's annotations, while the second term trains the expert models on observed annotations.*

## 4  Constraining the number of assignments in the E-step

A naive implementation of the EM algorithm often results in a workload-imbalance, where one or a few experts are queried more frequently, while the remaining experts remaining idle. Due to this nature, a naively-trained gating model would become biased toward the high-performing experts, resulting in a sub-optimal workload distribution among: (i) human experts themselves and (ii) human experts and the AI model, as mentioned in Section 1.

Ideally, in many applications where human experts are equally-competent, a perfect balanced workload among experts and the AI model can be expressed as follows:

$$
\tfrac{1}{N} \sum_{i=1}^{N} \mathbb{E}_{\widetilde{q}(\mathbf{z}_i)}[\mathbf{z}] = \tfrac{1}{N} \sum_{i=1}^{N} \widetilde{q}(\mathbf{z}_i) = \tfrac{1}{M+1} \mathbf{1},
\tag{6}
$$

where $\widetilde{q}$ is used to denote a constrained posterior to distinguish from the unconstrained posterior $q$ presented in Section 3. Note that this step happens between the E and M steps.

The workload balancing constraint in Eq. (6) is a special case, where all experts (including human and the AI classifier) are treated equally. To further generalise the control of the workload distributed to each expert, we propose the following workload constraint:

$$
\boldsymbol{\varepsilon}_l \preceq \tfrac{1}{N} \sum_{i=1}^{N} \widetilde{q}(\mathbf{z}_i) \preceq \boldsymbol{\varepsilon}_u,
\tag{7}
$$

where $\boldsymbol{\varepsilon}_u, \boldsymbol{\varepsilon}_l \in [0, 1]^{M+1}$ are hyper-parameters, and $\preceq$ is the element-wise operator.

The constrained optimisation is integrated into the E step when all annotations are observed (standard L2D) or considered as a "second E step" when annotations are partially observed as presented in Section 3. Nevertheless, the constrained optimisation can be written as follows:

$$
\widetilde{q}_i^* = \operatorname{argmin}_{\widetilde{q}} \mathrm{KL}\left[\widetilde{q}(\mathbf{z}_i) \| q^*(\mathbf{z}_i)\right], \forall i \in \{1, \ldots, N\} \quad \text{s.t.:} \quad \boldsymbol{\varepsilon}_l \preceq \tfrac{1}{N} \sum_{i=1}^{N} \widetilde{q}(\mathbf{z}_i) \preceq \boldsymbol{\varepsilon}_u,
\tag{8}
$$

where $q^*(\mathbf{z}_i)$ denotes the unconstrained posterior of $\mathbf{z}$, and can be either the true posterior $\Pr\left(\mathbf{z}_i \Big| \mathbf{x}_i, \mathbf{y}_i, \{\mathbf{t}_i^{(m)}\}_{m=1}^{M+1}, \{\theta_m\}_{m=1}^{M+1}, \gamma\right)$ in case of full-observed annotations (standard L2D) or the approximate posterior $q(\mathbf{z})$ obtained in Eq. (4).

The constrained optimisation in Eq. (8) is also known as the I-projection in information geometry and can be solved efficiently through its duality (Graça et al., 2007). Readers are referred to Appendix C for the detailed derivation of the solution. In general, the Lagrange multipliers obtained through the duality can be written as follows:

$$
\boldsymbol{\lambda}_u^*, \boldsymbol{\lambda}_l^* = \operatorname{argmin}_{\boldsymbol{\lambda}_u, \boldsymbol{\lambda}_u \geq 0} \boldsymbol{\lambda}_u^\top \boldsymbol{\varepsilon}_u - \boldsymbol{\lambda}_l^\top \boldsymbol{\varepsilon}_l + \tfrac{1}{N} \sum_{i=1}^{N} \ln \sum_{\mathbf{z}_i} q^*(\mathbf{z}_i) \exp\left(-(\boldsymbol{\lambda}_u - \boldsymbol{\lambda}_l)^\top \mathbf{z}_i - 1\right).
\tag{9}
$$

This can be solved efficiently using the projected gradient solver. The constrained posterior of the latent variable can then be obtained as follows:

$$
\widetilde{q}^*(\mathbf{z}_i) \propto q^*(\mathbf{z}_i) \exp\left(-(\boldsymbol{\lambda}_u^* - \boldsymbol{\lambda}_l^*)^\top \mathbf{z}_i - 1\right).
\tag{10}
$$

Please refer to Appendix D for a detailed training procedure of our proposed method.

The controllable workload constraint proposed in this section not only allows to distribute the workload more flexibly, but also facilitate the analysis of L2D, especially the *accuracy - coverage curve* (see Subsection 5.2). Current practice when analysing accuracy - coverage is to either:

- vary the cost of selection for human experts and classifier (Narasimhan et al., 2022), or
- add another loss term in the M step: $L = L_{M-step} + \zeta(Pr_{M+1}(\mathbf{z}|\mathbf{x};\gamma) - c)^2$, where $c$ is the desired coverage value and $\zeta$ is a weighting hyper-parameter, or
- use the prediction values from the gating model (Mozannar et al., 2023) to rank the test samples, and then plot the accuracy-coverage curve by adjusting the prediction threshold in a post-hoc fashion. (see Algorithm 2 in Appendix E).

The first two approaches above often require multiple attempts to fine-tune the cost per human expert or the hyper-parameter $\zeta$ associated with the new loss, resulting in a time-consuming and resource-wasting training procedure. In our experiments, we were unable to successfully train for some coverage values. In fact, we empirically observed that both of the approaches to constrain workload distribution are highly sensitive to pre-defined coverage values $c$, resulting in notably brittle performance. For example, when running on Cifar-100 as described in Section 5 using the second method to constrain the coverage at $c = 0.2$, we observed that setting $\zeta = 1$ leads to zero coverage, while setting $\zeta = 50$ leads to a coverage of 1. Applying a strategy similar to the bisection method to gradually narrow the range of $\zeta$ did not lead to successful convergence during training.

The third approach of threshold adjusting for post-hoc plotting (Mozannar et al., 2023) requires human intervention after each test sample has been processed by the system. Hence, it is inconsistent with the L2D training, resulting in an unreliable evaluation. Furthermore, it needs to know the gating model estimates for all of the testing samples before performing deferral, which is impractical, especially for model deployment.

In contrast, our proposed workload constrained optimisation in the E step is effective in terms of distributing the workload both among experts and between experts and the AI model without fine-tuning any hyper-parameter. In addition, it results in a train-test consistent L2D model, where the gating model makes decision at test time without post-hoc human intervention.

## 5 EXPERIMENTS

We evaluate our proposed method and the state-of-the-art methods on both synthetic (i.e., Cifar-100 (Krizhevsky, 2009)) and real-world (i.e., NIH-ChestXray (Majkowska et al., 2020), Chaoyang (Zhu et al., 2022) and MiceBone) datasets to benchmark L2D approaches.

For Cifar-100, we follow a similar setting as in (Hemmer et al., 2023), which simulates two experts, each following an asymmetric label noise. In particular, each expert performs correctly on 10 super-classes, while making 50 percent of labelling mistakes on the remaining 10 super-classes.

For real-world datasets, we use the annotations made by the human experts associated with the datasets. For each dataset, we randomly remove the annotations made by each human expert at a predefined missing rate to form the dataset with missing expert's annotations. In terms of human experts, NIH-AO dataset has one expert who performs at 99% prediction accuracy; Chaoyang dataset has three experts, with one performing at 91% accuracy, the other at 87%, and the last one at 99%; while MiceBone has eight experts, each performing at approximately 80% to 85% prediction accuracy. Note that for Chaoyang, we evaluate on two settings: one with all three experts and the other with the first two experts (excluding the best expert with 99% accuracy).

All results are reported from the checkpoint obtained at the last iteration in each training. Please refer to Appendix F for further details on the datasets and hyper-parameters used. The full implementation can be found at `https://github.com/cnguyen10/pl2d`.

### 5.1 BASELINES

Three baselines are used in the L2D evaluation with missing expert's annotations: "naive" (Madras et al., 2018) and two variants of semi-supervised learning (SSL) (Hemmer et al., 2023).

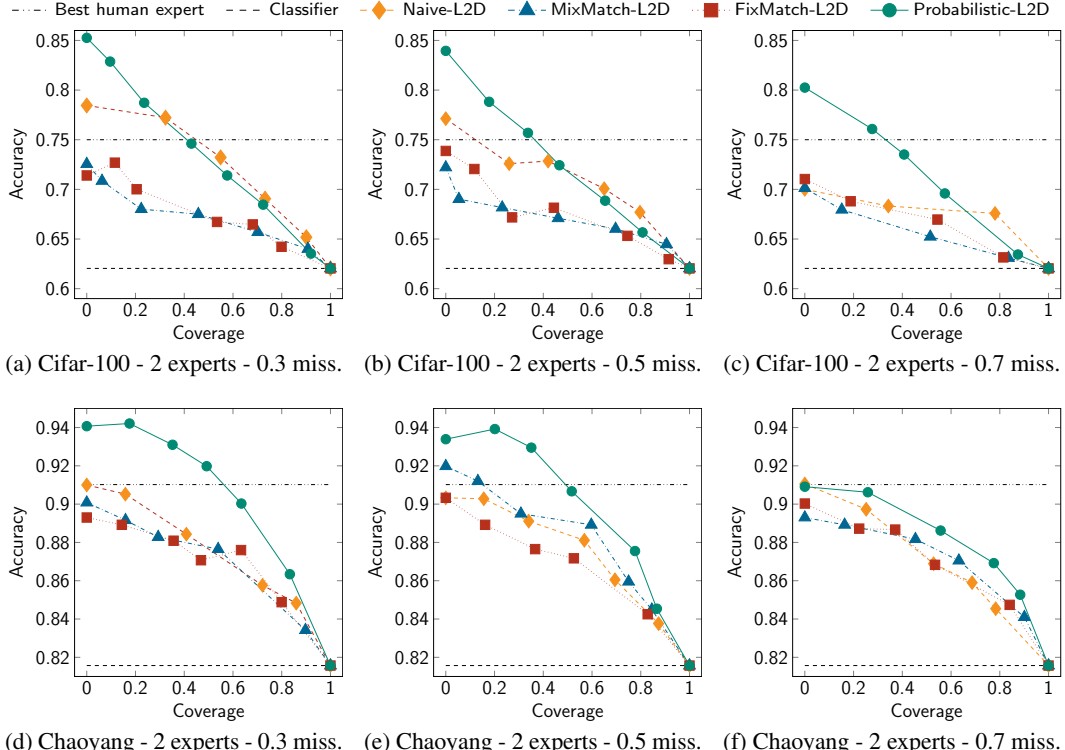

Figure 3: Comparison of coverage - accuracy curves between different L2D methods on a variety of datasets, each at a different missing rate.

The naive baseline is built from the original L2D method in (Madras et al., 2018), but with the gating model being modified to repeatedly sample either a human or AI classifier until the selected one is able to provide an annotation for that input data. Such a selection is equivalent to setting the probability $\Pr(\mathbf{z}|\mathbf{x}, \gamma)$ of the missing experts to zero in our EM framework (see Appendix A.2).

For the SSL baselines (Hemmer et al., 2023), each expert is modelled as an independent SSL classifier, where labelled data is represented by samples with observed annotations, while un-labelled data is the one with missing annotations. Although different SSL methods can be used to train each classifier to model each expert, for simplicity, we use MixMatch (Berthelot et al., 2019) and FixMatch (Sohn et al., 2020) because of their well-documented effectiveness in dealing with SSL problems. Missing annotations are then replaced by labels predicted by the SSL classifier. Finally, the standard L2D method (Madras et al., 2018) with the integration of workload constraint presented in Section 4 is trained on the "pseudo-complete" annotation dataset.

Note that although other L2D methods that handle multiple experts can be used (e.g., multi-L2D (Verma et al., 2023)), they do not provide a mechanism to control the workload distribution to study the trade-off in the coverage-accuracy-curves, and hence, are omitted from our evaluation.

## 5.2 RESULTS

Because L2D trades off the reliability of human experts for the efficiency of AI classifier, such a tradeoff is, therefore, analysed through the *coverage - accuracy* curve. *Coverage* computes the percentage of test samples assigned and predicted by the AI classifier, while *accuracy* measures the final prediction of the L2D system.

Fig. 3 shows the *coverage-accuracy* curves of the baselines and our proposed method on various datasets, each at different missing rates. In Cifar-100, our proposed probabilistic L2D outperforms the baselines at small coverage, while being on par at larger coverage values. The SSL baseline performs poorly because the synthetic annotations (i.e., randomly flipping labels between two classes) do not always satisfy the smooth manifold assumption in which similar samples have similar annotations. In Chaoyang and NIH-AO (shown in Appendix G.1) datasets, the baselines perform similarly while our method slightly surpasses them. In MiceBone, all methods perform worse than the best

Table 1: Area-under-coverage-accuracy-curve ($\times 100$) of different learning to defer methods when dealing with missing annotations on different datasets (larger is better). Note that at no missing annotation (or zero missing rate), all methods are simplified to the same learning algorithm, and hence, share the same result.

| Dataset | № experts | Missing rate | Naive L2D | MixMatch | FixMatch | Probabilistic (ours) |
|---------|-----------|--------------|-----------|----------|----------|----------------------|
| Cifar-100 | 2 | 0 | 75.38 | 75.38 | 75.38 | 75.38 |
|  |  | 0.3 | 72.80 | 66.95 | 67.39 | **73.33** |
|  |  | 0.5 | 70.83 | 66.80 | 67.27 | **72.13** |
|  |  | 0.7 | 67.68 | 65.52 | 66.52 | **71.19** |
| NIH-AO | 1 | 0.2 | 91.79 | 92.26 | 91.41 | **93.40** |
| Chaoyang | 2 | 0 | 89.49 | 89.49 | 89.49 | 89.49 |
|  |  | 0.3 | 87.48 | 86.90 | 86.86 | **90.50** |
|  |  | 0.5 | 87.46 | 88.12 | 86.66 | **90.03** |
|  |  | 0.7 | 87.09 | 87.11 | 86.92 | **88.41** |
|  | 3 | 0.5 | 89.94 | 88.55 | 90.33 | **91.07** |
| MiceBone | 8 | 0 | 79.58 | 79.58 | 79.58 | 79.58 |
|  |  | 0.3 | 79.67 | 81.02 | 81.87 | **83.48** |
|  |  | 0.5 | 80.30 | 82.19 | 82.12 | **83.69** |
|  |  | 0.7 | 81.31 | 80.60 | 82.55 | **83.10** |

human expert at small coverage. This is due to the slightly inconsistent performance of human experts in training and testing sets (see Table 2 in Appendix F). Nevertheless, our approach shows relatively better results, particularly at low coverage.

For a further comparison, we calculate the *area-under-coverage-accuracy-curve* (AUC-CAC) of each coverage - accuracy curve in Fig. 3 and the additional results in Appendix G.1, and report in Table 1. In general, our proposed probabilistic L2D outperforms the baselines across all datasets at various missing rates.

In terms of missing expert's annotation rates (see Fig. 6 in Appendix G.2 for further results of coverage-accuracy curves), the results on Cifar-100 in Table 1 agrees with our intuition, in which the higher the missing rate, the lower the AUC-CAC. This is slightly different in Chaoyang and MiceBone datasets, especially MiceBone at lower missing expert's annotation rates. Such a difference is mainly attributed to the inconsistent performance in training and testing sets (see Table 2 in Appendix F) coupled with missing data. For example, without any regularisation, training a naive L2D baseline on the MiceBone dataset would prefer the human expert `id 534` when there is no missing data, while selecting expert `id 290` when 70 percent of data is missing due to their highest performance in each setting (see the rows highlighted in Table 2). Such selections, coupled with their inconsistent performance between training and testing, lead to the observation: the system performs better at higher missing rate.

### 5.3 ABLATION STUDY ON CONTROLLABLE WORKLOAD

To study the controllable workload further, we perform additional experiments using the proposed probabilistic L2D under two settings: 1) *imbalanced* in which the workload of each human expert is not controlled, and 2) *balanced* in which the workload is divided evenly for all human experts. It is equivalent to setting $\varepsilon_l = 0$ and $\varepsilon_u = 1$ in the former configuration, while setting $\varepsilon_l \approx \varepsilon_l = (1 - \text{coverage})/M$ for each human expert in the latter one.

Figs. 4a and 4b shows the results on Chaoyang dataset with 2 and 3 human experts, respectively. At the same coverage, the *imbalanced*-workload approach performs slightly better than the *balanced* one. However, it is clear that the *imbalanced* one is biased toward the human expert with highest performance – note expert 1 in Fig. 4a and expert 3 in Fig. 4b. This bias is more severe in Fig. 4b where almost half of all testing samples are deferred to the expert 3, who has 99% prediction accuracy.

Fig. 4c shows another disadvantage of the *imbalanced* approach on MiceBone – a dataset with inconsistent human performance in the training and testing sets (see Table 2 in Appendix F). Without balancing the workload, L2D tends to prioritise the human expert with the highest performance in the training set, resulting in an overfitting. In contrast, the *balanced* approach regularises the training, and hence, leads to higher performance overall.

Nevertheless, the gain of overloading to one expert (or imbalanced approach) is small (less than 1 percent prediction accuracy in the Chaoyang dataset) or even negative (in MiceBone), and hence,

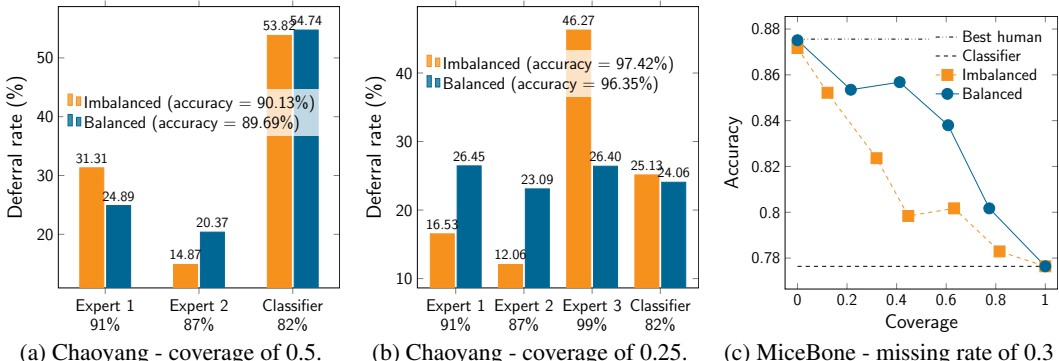

Figure 4: (a) and (b) show comparisons of two different workload constraints on Chaoyang dataset with 50% missing annotations per expert, where in the *imbalanced* setting, $\varepsilon_l = 0$ and $\varepsilon_u = 1$ for each human expert, while in the *balanced* setting, $\varepsilon_l \approx \varepsilon_u = (1 - \text{coverage})/M$ for each human expert, and (c) coverage - accuracy curve on MiceBone at 30% missing rate.

may not compensate for overworking the best expert. In fact, by overworking that expert, it is hard to guarantee that the expert's accuracy will not be affected. In practice, it is, therefore, preferred to distribute the workload evenly across all human experts.

## 6 RELATED WORK

**Learning to defer** extends the capability of *learning to reject* (Chow, 1957) by integrating human experts into the decision making process (Madras et al., 2018). In its simplest form, L2D consists of a classifier and a gating model that analyses an input sample and queries an expert (either a human or the classifier) to predict the label of that sample. By complementing the strengths of human experts and the classifier, L2D can perform better than the human experts or the classifier in isolation at a lower cost compared with if all samples need to be analysed by a human expert. Most studies in the L2D literature focuses on statistical learning that develops surrogate losses for the 0-1 loss to make the learning consistent with the optimal Bayes classifier and gating model (Mozannar & Sontag, 2020) or well-calibrated (Verma & Nalisnick, 2022). In addition, most studies consider only the setting of a single human expert, while in practice, there may be several human experts available. There are a few studies that considers the practical setting of multiple experts by either revising the gating model to output a weight for each expert (Keswani et al., 2021), using mixture of experts to produce a weighted average over the predictions of human experts and AI classifier (Hemmer et al., 2022), or extending further the surrogate loss (Verma et al., 2023). In contrast, our proposed method follows a probabilistic approach by modelling L2D as a variant of mixture of experts, and can be optimised by employing the EM algorithm. In addition, given that it is based on mixture of experts, our method can be straightforwardly extended to multiple human experts. The main difference between (Keswani et al., 2021; Verma et al., 2023) and our method, beside the modelling, is the codomain of the gating model. In our method, the output space of the gating model is the probability simplex $\Delta_M := \{\mathbf{y} : \mathbf{y} \in [0,1]^{M+1} \wedge \mathbf{y}\mathbf{1}^\top = 1\}$, meaning that the probability of deferring to one expert is influenced by the probability of deferring to other experts. In (Keswani et al., 2021), the codomain of the gating model is the hypercube $[0,1]^{M+1}$, meaning that the deferral probability of each expert is independent. In (Verma et al., 2023), the codomain is $\mathbb{R}^{C+M+1}$ due to the unification of the classifier and gating model into a single model.

**Missing experts' annotations** is one of the challenges in L2D (Leitão et al., 2022) because the standard L2D setting requires that every expert must annotate the whole training dataset. This is impractical because that increases the annotation cost significantly, or even infeasible for large-scale datasets. To our best knowledge, there is one study in the literature addressing such an issue (Hemmer et al., 2023). That study proposes a two-phase method: (i) train a semi-supervised learning model for each expert where the observed annotations are represented by the labels and the missing ones are the unlabelled samples, then (ii) use the synthesised pseudo-labels to replace the missing annotations and obtain a "complete" annotation dataset to train an L2D system. Due to the nature of two phase training, an error in the semi-supervised learning phase may be amplified in the training of L2D systems, potentially deteriorating performance. In contrast, our method models the missing

annotations to train a L2D system in an end-to-end manner. Such a modelling avoids error due to separate training, resulting in a high-performing and efficient training.

# 7 LIMITATIONS AND DISCUSSION

**Scalability** The proposed method, however, has a limited scalability with the number of human experts. As the number of human experts grows, the number of models required to represent them also increases, leading to a substantial rise in GPU memory and computational time needed for training (e.g, see Table 3). Although such an issue can be worked around by distributing each model of human expert to a different GPU, it is expensive due to the need of multiple GPUs. Thus, future work will focus on addressing the scalability of the proposed method, such as clustering $M$ experts into $K$ groups ($M \gg K$), where experts in the same group perform similarly. In that case, we only need $K$ models to represent the experts in each group, making it feasible to work with datasets consisting of hundreds or even thousands of human experts. We can also apply further dimensional reduction techniques, such as hierarchical clustering, if the labelling pattern of human experts is diverse. Alternatively, one can design a conditional model, denoted as $h(\mathbf{x}, \zeta)$ with $\zeta$ being the embedding of a human expert, to model each human expert. The embedding $\zeta$ can be obtained by extending from sample-wise to set-wise representation learning through the usage of *deep set* (Zaheer et al., 2017). Such a conditional modelling can adapt to any number of human experts, and hence, addresses the scalability issue.

**Deferral at low coverage** As we mentioned in Section 1, L2D aims to maximise reliability, while keeping the cost within the acceptable limits. The choice of the ideal operating point that balances reliability and costs typically depends on specific priorities. In high-stakes applications (e.g., healthcare), deferring uncertain cases offers significant advantages, such as improving decision accuracy, enhancing trust by acknowledging the limitations of the classifier, and leveraging human oversight to ensure accountability in high-stakes scenarios. Deferred cases can also provide valuable learning opportunities to improve the AI model. However, deferring cases can be costly due to increased annotation budgets, potential expert burnout, and possible delays in decision-making. On the other hand, minimising deferrals maximises the efficiency of the L2D system, allowing for lower costs and quicker decision-making. This approach also reduces reliance on human experts and demonstrates the capability of the classifier to handle easier cases independently. However, not deferring uncertain cases poses risks, such as potential misdiagnoses and diminished oversight. These risks can have severe consequences for patient safety and undermine trust in the system, especially when high-stakes decisions are involved.

**Dynamic expert performance** can be accommodated by incorporating temporal dynamics and performance variability through sequential modelling techniques or adaptive mechanisms. For fast-changing performance, such as fatigue, the model could include real-time performance tracking using metrics such as dynamic accuracy, allowing the adjustment of the deferral strategy based on current conditions. Slow-changing performance, like expertise improvement through learning, can be modelled using techniques such as Bayesian updating to adapt expert-specific parameters over time. Temporal modelling (e.g., recurrent neural networks) can also capture patterns in performance changes, enabling the system to make adjustments. By integrating such adaptive mechanisms, the model would better reflect the evolving nature of human expertise, improving its decision-making capabilities in dynamic scenarios.

# 8 CONCLUSION

This paper proposes a probabilistic modelling for L2D that extends the capability of the standard L2D to a practical setting where each human expert does not need to annotate the whole dataset of interest, but only a portion of the dataset. By modelling the expert selection and missing human experts' annotations as latent variables, the proposed method is optimised through the EM algorithm. We also propose to integrate a workload control mechanism as a constrained optimisation in the E step to enable the capability of distributing the workload to human experts and the classifier efficiently. Empirical evaluation demonstrates that the proposed probabilistic L2D outperforms prior methods in both synthetic and real-world datasets with different missing rates of annotations.

ACKNOWLEDGEMENT

This research was supported by the Engineering and Physical Sciences Research Council (EPSRC) - UK Research and Innovation through the grant EP/Y018036/1. We also thank Dr David Rosewarne from Royal Wolverhampton Hospitals NHS Trust for providing useful discussion on the point of view of radiologists to further motivate this study.

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

## A  BASELINE OF LEARNING TO DEFER

### A.1  FULL DATA ANNOTATIONS

This follows the standard setting of learning to defer, in which every sample is annotated by all experts. In this subsection, we present the EM algorithm to optimise the objective function in Eq. (2).

The "complete" data log-likelihood in this case can be written as follows:

$$
\begin{aligned}
Q_i(\gamma^{(k)}, \gamma, \theta_{M+1}) = \mathbb{E}_{\Pr\left(\mathbf{z}_i | \mathbf{x}_i, \mathbf{y}_i, \prod_{m=1}^{M+1} \mathbf{t}_i^{(m)}, \gamma^{(k)}\right)} & \left[ \ln \Pr\left( \mathbf{y}_i \middle| \mathbf{z}_i, \prod_{m=1}^{M+1} \mathbf{t}_i^{(m)} \right) + \ln \Pr(\mathbf{z}_i | \mathbf{x}_i, \gamma) \right] \\
& + \ln \Pr(\mathbf{t}_i = \mathbf{y}_i | \mathbf{x}_i, \theta_{M+1}).
\end{aligned}
$$
(11)

**E step**  calculates the posterior of the expert index $\mathbf{z}$ as follows:

$$
\Pr\left( \mathbf{z}_i \middle| \mathbf{x}_i, \mathbf{y}_i, \prod_{m=1}^{M+1} \mathbf{t}_i^{(m)} \right) \propto \Pr\left( \mathbf{y}_i \middle| \mathbf{z}_i, \prod_{m=1}^{M+1} \mathbf{t}_i^{(m)} \right) \Pr(\mathbf{z}_i | \mathbf{x}_i, \gamma).
$$
(12)

**M step**  maximises the complete-data log-likelihood as follows:

$$
\gamma^{(k+1)}, \theta_{M+1} \leftarrow \operatorname*{argmax}_{\gamma, \theta_{M+1}} \sum_{i=1}^{N} Q_i(\gamma^{(k)}, \gamma, \theta_{M+1}).
$$
(13)

As mentioned in Section 4, one can apply the constrained E step right after the E step to obtain a constrained posterior on $\mathbf{z}$ before performing the maximisation in the M step.

### A.2  MISSING EXPERT'S ANNOTATIONS

In this subsection, every sample is no longer annotated by all experts, but at least one expert. The objective function can be written as follows:

$$
\begin{aligned}
& \max_{\gamma, \theta_{M+1}} \sum_{i=1}^{N} \ln \Pr\left( \mathbf{y}_i, \prod_{m \in \mathcal{D}_i^{\text{obs.}}} \mathbf{t}_i^{(m)} \middle| \mathbf{x}_i, \gamma, \theta_{M+1} \right) \\
& = \max_{\gamma, \theta_{M+1}} \sum_{i=1}^{N} \ln \Pr\left( \mathbf{y}_i \middle| \mathbf{x}_i, \prod_{m \in \mathcal{D}_i^{\text{obs.}}} \mathbf{t}_i^{(m)}, \gamma \right) + \ln \Pr\left( \mathbf{t}_i^{(M+1)} | \mathbf{x}_i, \theta_{M+1} \right).
\end{aligned}
$$
(14)

For the naive learning to defer baseline, we sample an expert from $\Pr(\mathbf{z} | \mathbf{x}, \gamma)$ until the sampled expert has an annotation available for the sample of interest. This is equivalent to:

$$
\Pr(\mathbf{z}_i = j | \mathbf{x}_i, \gamma) = 0, \forall j \in \mathcal{D}_i^{\text{unobs.}}.
$$
(15)

We then follow the exact EM algorithm specified in Appendix A.1.

## B  UNCONSTRAINED POSTERIOR OF LATENT VARIABLES IN THE E STEP

We employ the variational EM to optimise the objective function in Eq. (2). The variational inference is applied in the E step to approximate the posterior of the latent variables $\mathbf{z}$ and $\mathbf{t}$. The variational distribution can be written as follows:

$$
q(\mathbf{z}, \prod_{j \in \mathcal{D}^{unobs.}} \mathbf{t}^{(j)}) = \underbrace{q(\mathbf{z}; \rho)}_{\text{Categorical}(\mathbf{z}; \rho)} \prod_{j \in \mathcal{D}^{\text{unobs.}}} \underbrace{q(\mathbf{t}^{(j)}; \phi^{(j)})}_{\text{Categorical}(\mathbf{t}^{(j)}; \phi^{(j)})},
$$
(16)

where the variational parameters $\rho$ and $\phi^{(j)}$ are probability vector and obtained through the optimisation in Eq. (4).

The optimisation in the "unconstrained" E step is restated here to ease the derivation:

$$
\underset{q}{\arg\min} \, \mathrm{KL}\left[ q(\mathbf{z}_i; \rho_i) \| \Pr\left( \mathbf{z}_i | \mathbf{x}_i, \gamma^{(k)} \right) \right] + \sum_{j \in \mathcal{D}_i^{\mathrm{unobs.}}} \mathrm{KL}\left[ q(\mathbf{t}_i^{(j)}; \phi_i^{(j)}) \, \Big\| \, \Pr\left( \mathbf{t}_i^{(j)} \Big| \mathbf{x}_i, \theta_j^{(k)} \right) \right]
$$
$$
- \mathbb{E}_{q(\mathbf{z}_i; \rho_i) \prod_{j \in \mathcal{D}_i^{\mathrm{unobs.}}} q\left( \mathbf{t}_i^{(j)}; \phi_i^{(j)} \right)} \left[ \ln \Pr\left( \mathbf{y}_i \Big| \mathbf{z}_i, \prod_{m \in \mathcal{D}_i^{\mathrm{obs.}}} \mathbf{t}_i^{(m)}, \prod_{j \in \mathcal{D}_i^{\mathrm{unobs.}}} \mathbf{t}_i^{(j)} \right) \right]. \tag{4}
$$

The objective function in Eq. (4) (also known as *variational-free energy*) can be expanded as follows:

$$
\mathsf{L}_{\mathrm{vfe}} = \sum_{n=1}^{M+1} \rho_{in} \left[ \ln \rho_{in} - \ln g_n(\mathbf{x}_i; \gamma^{(k)})) \right] + \sum_{j \in \mathcal{D}_i^{\mathrm{unobs.}}} \sum_{c=1}^{C} \phi_{ic}^{(j)} \left[ \ln \phi_{ic}^{(j)} - \ln f_c(\mathbf{x}_i; \theta_j^{(k)}) \right]
$$
$$
- \sum_{n=1}^{M+1} \sum_{c=1}^{C} \rho_{in} \prod_{j \in \mathcal{D}^{\mathrm{unobs.}}} \phi_{ic}^{(j)} \mathbf{e}_{\mathbf{y}_i} \ln \zeta_{i\mathbf{y}_i}^{(n)}, \tag{17}
$$

where: $\mathbf{e}_v$ is the $v$-th unit vector (often known as one-hot vector), and $\zeta_{ij}^{(n)}$ is the $j$-th element of $\zeta_i^{(n)}$ defined as follows:

$$
\zeta_i^{(n)} = \begin{cases} \mathbf{t}_i^{(n)} & \text{if } n \in \mathcal{D}^{\mathrm{obs.}} \\ f(\mathbf{x}_i; \theta_n) & \text{if } n \in \mathcal{D}^{\mathrm{unobs.}} \end{cases}. \tag{18}
$$

**Variational parameter $\rho$**  We minimise the variational-free energy in Eq. (17) w.r.t. $\rho$, which is the probability to select which expert (either human or the classifier) among a total of $M+1$ experts. Note that because $\rho$ is a probability, it is constrained by $\sum_{n=1}^{M+1} \rho_{in} = 1$.

We form the Lagrangian by isolating the terms which contain $\rho_i$ and adding the appropriate Lagrange multiplier, $\eta_i \geq 0$, as follows:

$$
\mathsf{L}_{\mathrm{vfe}}[\rho_i] = \sum_{n=1}^{M+1} \rho_{in} \left[ \ln \rho_{in} - \ln g_n(\mathbf{x}_i; \gamma^{(k)})) \right] - \sum_{n=1}^{M+1} \sum_{c=1}^{C} \rho_{in} \prod_{j \in \mathcal{D}^{\mathrm{unobs.}}} \phi_{ic}^{(j)} \mathbf{e}_{\mathbf{y}_i} \ln \zeta_{i\mathbf{y}_i}^{(n)}
$$
$$
+ \eta_i \left( \sum_{n=1}^{M+1} \rho_{in} - 1 \right). \tag{19}
$$

Taking the derivative with respect to $\rho_{in}$ gives:

$$
\frac{\mathrm{d}\mathsf{L}_{\mathrm{vfe}}[\rho_i]}{\mathrm{d}\rho_{in}} = \ln \rho_{in} - \ln g_n(\mathbf{x}_i; \gamma^{(k)}) + 1 - \sum_{c=1}^{C} \prod_{j \in \mathcal{D}^{\mathrm{unobs.}}} \phi_{ic}^{(j)} \mathbf{e}_{\mathbf{y}_i} \ln \zeta_{i\mathbf{y}_i}^{(n)} + \eta_i. \tag{20}
$$

Setting the derivative to zero yields the optimal value of the variational paramter:

$$
\rho_{in} \propto g_n(\mathbf{x}_i; \gamma^{(k)}) \exp\left[ \sum_{c=1}^{C} \prod_{j \in \mathcal{D}^{\mathrm{unobs.}}} \phi_{ic}^{(j)} \mathbf{e}_{\mathbf{y}_i} \ln \zeta_{i\mathbf{y}_i}^{(n)} - 1 \right]. \tag{21}
$$

It can also be rewritten as follows:

$$
q(\mathbf{z}_i) \propto \Pr\left( \mathbf{z}_i | \mathbf{x}_i, \gamma^{(k)} \right)
$$
$$
\times \exp\left\{ \mathbb{E}_{\prod_{j \in \mathcal{D}_i^{\mathrm{unobs.}}} q\left( \mathbf{t}_i^{(j)} \right)} \left[ \ln \Pr\left( \mathbf{y}_i \Big| \mathbf{z}_i, \prod_{m \in \mathcal{D}_i^{\mathrm{obs.}}} \mathbf{t}_i^{(m)}, \prod_{j \in \mathcal{D}_i^{\mathrm{unobs.}}} \mathbf{t}_i^{(j)} \right) \right] - 1 \right\}. \tag{22}
$$

**Variational parameter** $\phi$   Similarly, we can obtain the variational parameter $\phi$ and write the solution in the following form:

$$q(\mathbf{t}_i^{(j)}) \propto \Pr\left(\mathbf{t}_i^{(j)}|\mathbf{x}_i, \theta_j^{(k)}\right)$$

$$\times \exp\left\{\mathbb{E}_{q(\mathbf{z}_i) \prod_{j' \in \mathcal{D}_i^{\text{unobs.}} \backslash j} q\left(\mathbf{t}_i^{(j')}\right)} \left[\ln \Pr\left(\mathbf{y}_i \,\Big|\, \mathbf{z}_i, \prod_{m \in \mathcal{D}_i^{\text{obs.}}} \mathbf{t}_i^{(m)}, \prod_{j \in \mathcal{D}_i^{\text{unobs.}}} \mathbf{t}_i^{(j)}\right)\right] - 1\right\}.$$

(23)

Eqs. (22) and (23) result in a system of $(M + 1 + C|\mathcal{D}_i|^{\text{unobs.}})$ non-linear equations in the form $x = f(x)$, which can be solved by *fixed-point* iteration method or the Newton - Raphson method. For simplicity, we use the fixed-point method with a fixed number of iterations as 10 to solve the above system of non-linear equations.

## C   CONSTRAINED POSTERIOR OF LATENT VARIABLES IN THE E STEP

This section provides the detailed derivation of the constrained optimisation in Eq. (8), which can be rewritten in the form of standard optimisation as follows:

$$\{\widetilde{q}_i^*\}_{i=1}^N = \underset{\{q_i\}_{i=1}^N}{\operatorname{argmin}} \frac{1}{N} \sum_{i=1}^N \mathbb{E}_{\widetilde{q}(\mathbf{z}_i)} \left[\ln \widetilde{q}(\mathbf{z}_i) - \ln q^*(\mathbf{z}_i)\right]$$

$$\text{s.t.:} \quad \frac{1}{N} \sum_{i=1}^N \widetilde{q}(\mathbf{z}_i) \preceq \boldsymbol{\varepsilon}_{\text{u}} \quad \wedge \quad -\frac{1}{N} \sum_{i=1}^N \widetilde{q}(\mathbf{z}_i) \preceq -\boldsymbol{\varepsilon}_{\text{l}}.$$

(24)

The Lagrangian of the constrained optimisation can be written as follows:

$$\mathsf{L} = \frac{1}{N} \sum_{i=1}^N \mathbb{E}_{\widetilde{q}(\mathbf{z}_i)} \left[\ln \widetilde{q}(\mathbf{z}_i) - \ln q^*(\mathbf{z}_i)\right] + \boldsymbol{\lambda}_{\text{u}}^\top \left[\left(\frac{1}{N} \sum_{i=1}^N \widetilde{q}(\mathbf{z}_i)\right) - \boldsymbol{\varepsilon}_{\text{u}}\right] + \boldsymbol{\lambda}_{\text{l}}^\top \left[\boldsymbol{\varepsilon}_{\text{l}} - \left(\frac{1}{N} \sum_{i=1}^N \widetilde{q}(\mathbf{z}_i)\right)\right]$$

$$= \frac{1}{N} \sum_{i=1}^N \mathbb{E}_{\widetilde{q}(\mathbf{z}_i)} \left[\ln \widetilde{q}(\mathbf{z}_i) - \ln q^*(\mathbf{z}_i)\right] + (\boldsymbol{\lambda}_{\text{u}} - \boldsymbol{\lambda}_{\text{l}})^\top \widetilde{q}(\mathbf{z}_i) - \boldsymbol{\lambda}_{\text{u}}^\top \boldsymbol{\varepsilon}_{\text{u}} + \boldsymbol{\lambda}_{\text{l}}^\top \boldsymbol{\varepsilon}_{\text{l}},$$

(25)

where $\boldsymbol{\lambda}_{\text{u}}, \boldsymbol{\lambda}_{\text{l}} \in \mathbb{R}_+^{M+1}$ are the Lagrange multipliers.

Taking the functional derivative (similar to the derivation of variational inference explained in David Blei's lecture: https://www.cs.princeton.edu/courses/archive/fall11/cos597C/lectures/variational-inference-i.pdf) with respect to $q(\mathbf{z}_i)$ gives:

$$\frac{\mathrm{d}\mathsf{L}}{\mathrm{d}\widetilde{q}_i} = \frac{1}{N} \left[\ln \widetilde{q}(\mathbf{z}_i) - \ln q^*(\mathbf{z}_i) + 1 + \boldsymbol{\lambda}_{\text{u}} - \boldsymbol{\lambda}_{\text{l}}\right]$$

(26)

Setting the KKT condition gives:

$$\frac{\mathrm{d}\mathsf{L}}{\mathrm{d}\widetilde{q}_i} = 0 \Leftrightarrow \ln \widetilde{q}(\mathbf{z}_i) - \ln q^*(\mathbf{z}_i) + \boldsymbol{\lambda}_{\text{u}} - \boldsymbol{\lambda}_{\text{l}} + 1 = 0.$$

(27)

Solving for $\widetilde{q}(\mathbf{z}_i)$ gives:

$$\ln \widetilde{q}(\mathbf{z}_i) \propto \ln q^*(\mathbf{z}_i) - \boldsymbol{\lambda}_{\text{u}} + \boldsymbol{\lambda}_{\text{l}} - 1.$$

(28)

Or:

$$\widetilde{q}(\mathbf{z}_i) = \frac{1}{Z(\boldsymbol{\lambda}_{\text{u}}, \boldsymbol{\lambda}_{\text{l}})} \frac{q^*(\mathbf{z}_i)}{\exp\left((\boldsymbol{\lambda}_{\text{u}} - \boldsymbol{\lambda}_{\text{l}})^\top \mathbf{z}_i + 1\right)},$$

(29)

where: $Z(\boldsymbol{\lambda}_{\text{u}}, \boldsymbol{\lambda}_{\text{l}})$ is the normalisation constant defined as follows:

$$Z(\boldsymbol{\lambda}_{\text{u}}, \boldsymbol{\lambda}_{\text{l}}) = \sum_{\mathbf{z}_i} \frac{q^*(\mathbf{z}_i)}{\exp\left((\boldsymbol{\lambda}_{\text{u}} - \boldsymbol{\lambda}_{\text{l}})^\top \mathbf{z}_i + 1\right)}.$$

(30)

---

**Algorithm 1** The EM algorithm for learning to defer with missing expert's annotations

---

1: **procedure** TRAINING($\mathcal{D}, \varepsilon_{\mathrm{l}}, \varepsilon_{\mathrm{u}}, n, K$)
2:     $\triangleright$ $\mathcal{D}_i$: training dataset defined in Section 2     $\triangleleft$
3:     $\triangleright$ $\varepsilon_{\mathrm{l}}, \varepsilon_{\mathrm{u}}$: the lower and upper workload constrained vectors     $\triangleleft$
4:     $\triangleright$ n: mini-batch size     $\triangleleft$
5:     $\triangleright$ K: number of iterations     $\triangleleft$
6:     initialise parameter of gating model $\gamma^{(0)}$
7:     initialise parameters of expert models $(\theta_m^{(0)})_{m=1}^{M+1}$
8:     **for** $k \in \{1, \ldots, K\}$ **do**
9:        draw a mini-batch of $n$ samples
10:        **for** $i \in \{1, \ldots, n\}$ **do**     $\triangleright$ Iterate through every sample in the mini-batch
11:          $q^*(\mathbf{z}_i), (q^*(\mathbf{t}_i^{(j)}))_{j \in \mathcal{D}^{\mathrm{unobs.}}} \leftarrow \min$ VARIATIONAL-FREE ENERGY $\triangleright$ E step in Eq. (4)
12:          $q^*(\mathbf{z}_i) \leftarrow$ CONSTRAINED POSTERIOR($\{q^*(\mathbf{z}_i)\}_{i=1}^n, \varepsilon_{\mathrm{l}}, \varepsilon_{\mathrm{u}}$)     $\triangleright$ Section 4
13:        calculate the data-completed log-likelihood $Q_i(\gamma, \theta, \gamma^{(k)}, \theta^{(k)})$     $\triangleright$ see Eq. (3)
14:        $\gamma^{(k+1)}, \theta^{(k+1)} \leftarrow \operatorname{argmax}_{\gamma, \theta} \sum_{i=1}^n Q_i(\gamma, \theta, \gamma^{(k)}, \theta^{(k)}) + \ln \Pr(\gamma) + \sum_{m=1}^{M+1} \ln \Pr(\theta_m)$
15:     **return** $\gamma, \theta_{M+1}$     $\triangleright$ Parameters of gating model and AI classifier

---

Substituting $\widetilde{q}(\mathbf{z}_i)$ in Eq. (29) into the Lagrangian in Eq. (25) gives:

$$\mathsf{L} = \frac{1}{N} \sum_{i=1}^N \mathbb{E}_{\widetilde{q}(\mathbf{z}_i)} \left[ -\boldsymbol{\lambda}_{\mathrm{u}} + \boldsymbol{\lambda}_{\mathrm{l}} - 1 - \ln Z(\boldsymbol{\lambda}_{\mathrm{u}}, \boldsymbol{\lambda}_{\mathrm{l}}) \right] + (\boldsymbol{\lambda}_{\mathrm{u}} - \boldsymbol{\lambda}_{\mathrm{l}})^\top \widetilde{q}(\mathbf{z}_i) - \boldsymbol{\lambda}_{\mathrm{u}}^\top \varepsilon_{\mathrm{u}} + \boldsymbol{\lambda}_{\mathrm{l}}^\top \varepsilon_{\mathrm{l}}$$

$$= -\boldsymbol{\lambda}_{\mathrm{u}}^\top \varepsilon_{\mathrm{u}} + \boldsymbol{\lambda}_{\mathrm{l}}^\top \varepsilon_{\mathrm{l}} - 1 - \frac{1}{N} \sum_{i=1}^N \ln Z(\boldsymbol{\lambda}_{\mathrm{u}}, \boldsymbol{\lambda}_{\mathrm{l}})$$

$$= -\boldsymbol{\lambda}_{\mathrm{u}}^\top \varepsilon_{\mathrm{u}} + \boldsymbol{\lambda}_{\mathrm{l}}^\top \varepsilon_{\mathrm{l}} - 1 - \frac{1}{N} \sum_{i=1}^N \ln \sum_{\mathbf{z}_i} \frac{q^*(\mathbf{z}_i)}{\exp\left((\boldsymbol{\lambda}_{\mathrm{u}} - \boldsymbol{\lambda}_{\mathrm{l}})^\top \mathbf{z}_i + 1\right)}. \tag{31}$$

According to the duality, $\boldsymbol{\lambda}_{\mathrm{u}}$ and $\boldsymbol{\lambda}_{\mathrm{l}}$ can be obtained through the following optimisation:

$$\boldsymbol{\lambda}_{\mathrm{u}}^*, \boldsymbol{\lambda}_{\mathrm{l}}^* = \operatorname*{argmax}_{\boldsymbol{\lambda}_{\mathrm{u}}, \boldsymbol{\lambda}_{\mathrm{u}} \geq 0} \mathsf{L} = \operatorname*{argmin}_{\boldsymbol{\lambda}_{\mathrm{u}}, \boldsymbol{\lambda}_{\mathrm{u}} \geq 0} \boldsymbol{\lambda}_{\mathrm{u}}^\top \varepsilon_{\mathrm{u}} - \boldsymbol{\lambda}_{\mathrm{l}}^\top \varepsilon_{\mathrm{l}} + \frac{1}{N} \sum_{i=1}^N \ln \sum_{\mathbf{z}_i} \frac{q^*(\mathbf{z}_i)}{\exp\left((\boldsymbol{\lambda}_{\mathrm{u}} - \boldsymbol{\lambda}_{\mathrm{l}})^\top \mathbf{z}_i + 1\right)}. \tag{32}$$

## D    TRAINING PROCEDURE

The training of the proposed probabilistic L2D with workload constraints is described in Algorithm 1.

## E    POST-HOC METHOD FOR ACCURACY - COVERAGE CURVES

In the standard learning to defer, the coverage constraint is imposed as a post-hoc during testing. The post-hoc procedure is shown in Algorithm 2 (refer to *helpers/metrics.py* in https://github.com/clinicalml/human_ai_deferral/ as the original code to implement the post-hoc algorithm). In particular, the confidence values produced by the gating model to select the AI classifier are used to sort the test samples, with the top-k samples being selected to defer to the AI classifier. Such a post-hoc testing procedure is inconsistent with the training procedure in learning to defer due to the human intervention. Furthermore, the procedure requires to access all of the testing samples ahead of time before performing deferral. This is, however, impractical, especially in model deployment.

---

**Algorithm 2** Current practice in calculating coverage and accuracy at test time

---

1: **procedure** COVERAGE-ACCURACY($\{(\mathbf{x}_i, (\mathbf{t}_i^{(m)})_{m=1}^M, \mathbf{y}_i)\}_{i=1}^N, \gamma, \theta, \tau$)
2: $\quad \triangleright \mathbf{x}_i$: *input sample* $\qquad\qquad\qquad\qquad\qquad\qquad\qquad\qquad\qquad\qquad\qquad\quad\triangleleft$
3: $\quad \triangleright \mathbf{t}_i^{(m)}$: *annotation made by expert indexed by* $m$ $\qquad\qquad\qquad\qquad\qquad\quad\triangleleft$
4: $\quad \triangleright \mathbf{y}_i$: *ground truth* $\qquad\qquad\qquad\qquad\qquad\qquad\qquad\qquad\qquad\qquad\qquad\quad\triangleleft$
5: $\quad \triangleright \gamma$: *parameter of gating (deferral) model* $\qquad\qquad\qquad\qquad\qquad\qquad\qquad\triangleleft$
6: $\quad \triangleright \theta$: *parameter of ML classifier* $\qquad\qquad\qquad\qquad\qquad\qquad\qquad\qquad\quad\triangleleft$
7: $\quad \triangleright \tau$: *coverage value of interest* $\qquad\qquad\qquad\qquad\qquad\qquad\qquad\qquad\quad\triangleleft$
8: $\quad$ initialise an emptied ordered set of confidence selecting the classifier: $\mathcal{P} \leftarrow ()$
9: $\quad$ **for** $i \in \{1, \ldots, N\}$ **do**
10: $\qquad \mathbf{p}_i \leftarrow g(\mathbf{x}_i; \gamma)$ $\qquad\qquad\qquad\qquad\qquad \triangleright$ *probability of routing/selection*
11: $\qquad \mathcal{P} \leftarrow \mathcal{P} \cup \mathbf{p}_{i,M+1}$ $\qquad\qquad\qquad \triangleright$ *store confidence of selecting the classifier*
12: $\quad \mathcal{I}_{\text{clf}} \leftarrow$ ARG-TOP-K(population $= \mathcal{P}, k = \lfloor \tau N \rfloor$) $\triangleright$ *Indices of samples classified by classifier*
13: $\quad \mathcal{I}_{\text{experts}} \leftarrow \{1, \ldots, N\} \backslash \mathcal{I}_{\text{clf}}$ $\quad \triangleright$ *Indices of samples classified by human experts or classifier*
14: $\quad$ **return** total accuracy

---

# F EXPERIMENT SETTING

## F.1 DATASETS

**Cifar-100** consists of 50,000 training and 10,000 testing images which are categorised into 20 superclasses. These superclasses are then divided further into 100 classes. We follow the approach of (Hemmer et al., 2023) to simulate two experts, each exhibiting asymmetric label noise. In particular, each expert classifies correctly on 10 designated superclasses, while making errors 50% of the time on the remaining 10 superclasses. The annotations made by each expert is then randomly sampled at a missing rate of interest. Moreover, given that the Cifar-100 dataset has about 10 percent of testing images that are duplicated or almost identical to the ones in the training set, we decide to use ciFAIR-100 (Barz & Denzler, 2020), which replaces those duplicated images by ones that belong to the same classes.

**NIH-ChestXray** consists of 4,374 chest X-ray images taken from the ChestXray-8 dataset (Wang et al., 2017). Each X-ray image is initially reviewed by three radiologists over four radio-graphic findings. If there is a disagreement between the three radiologists, the image is returned for an additional review. The labels and notes from the previous rounds are made available during each iterative review. Adjudication continues until a consensus is reached or for a maximum of five rounds to determine the ground truth label. Although there are a total of 22 radiologists (including board-certified and resident radiologists), each radiologist annotates only a subset of the whole dataset with small mutual overlapping, increasing the difficulty to form a test set for learning to defer. We follow a similar approach as in (Hemmer et al., 2023) to select the radiologist with id `4295342357`, who has the largest number of annotations – 2,350 images – for the evaluation. To form a train - test split, we randomly select 20% of patients to be in the test subset, while the remaining patients are used in the training subset. We further randomly remove 20% of the annotations from the training subset to simulate the missing annotations. In addition, the original image size of 1,024-by-1,024 is resized to 256-by-256 before randomly cropping to 224-by-224 during training.

**Chaoyang** consists of 4,021 training and 2,139 testing images of colon patches obtained from the Chaoyang hospital in China (Zhu et al., 2022). Each image is annotated by three professional pathologists over four categories: normal, serrated, adenocarcinoma, and adenoma. The ground truth labels are obtained via the majority vote of the three annotations. The performance of the three pathologists are 91%, 87% and 99% accurate according to this majority vote. In our experiments, we show results with all three experts, and with two experts, where we exclude the third expert.

**MiceBone** consists of 4,736 *second-harmonic generation microscopy* images from 35 3D-scans of 6 mice where 3 mice had the disease *osteogenesis imperfecta* – also known as brittle bone disease, while the others do not (Schmarje et al., 2019). We follow the same preprocessing procedure in (Schmarje et al., 2022b) by using the given segmentation masks to cut the original 2D image slices

into many patches, each consisting of one class, resulting in a total of 7,240 2D-images. Each image is then annotated by one to five professional annotators to classify into three classes (Schmarje et al., 2022a). Among the 79 annotators, only 8 annotators annotate the whole dataset. Hence, we use the annotations of those 8 annotators to represent 8 experts in our experiment, while using the majority vote as the ground truth. The performance of those 8 experts are shown in Table 2. For the train - test split, we use the first four folds as the training set, and the remaining fold as the test set.

Table 2: The performance of 8 experts from the MiceBone dataset.

| Expert id | Prediction accuracy (%) | | | | |
| --- | --- | --- | --- | --- | --- |
| | Train | (with missing rate) | | | Test |
| | 0 | 0.3 | 0.5 | 0.7 | |
| 047 | 86.34 | 64.45 | 49.15 | 34.68 | 85.74 |
| 290 | 85.82 | 65.63 | 49.82 | **36.81** | 85.29 |
| 533 | 86.29 | 64.37 | 49.61 | 35.30 | 84.96 |
| 534 | **86.59** | 64.47 | 50.11 | 35.00 | 84.45 |
| 580 | 80.76 | 60.40 | 47.90 | 33.79 | 79.07 |
| 581 | 85.83 | 64.12 | 50.43 | 36.37 | 84.32 |
| 966 | 86.54 | 64.47 | 49.41 | 35.62 | 87.56 |
| 745 | 85.01 | 65.61 | 50.71 | 35.81 | 84.38 |

### F.2 TRAINING HYPER-PARAMETERS

The model used throughout the evaluation is mainly Resnet-18 for datasets that have large size images (e.g., NIH, Chaoyang and MiceBone), while Pre-Act-Resnet-18 is used in datasets with smaller image size (e.g., ciFAIR-100). Each method is trained for 300 epochs (1,000 epochs for ciFAIR-100) using stochastic gradient descent with a momentum of 0.9 and a learning rate of 0.01. The learning rate is decayed through a cosine decaying scheduler, and the gradient norm is clipped at the maximal of 10 for numerical stability. In addition, a mixed precision using `bfloat16` is applied over all methods and datasets to speed up the training. The proposed method is implemented in Jax – a Python library that accelerates array computation and program transformation to achieve high-performance numerical computing for large-scale machine learning.

For the baseline relying on semi-supervised learning, we also use the same backbone. The trained model is then used to produce pseudo-labels for the missing annotations. A standard L2D is then trained on the observed labels and pseudo-labels to produce the final results.

## G ADDITIONAL RESULTS

### G.1 COVERAGE - ACCURACY CURVES ON NIH-AO, CHAOYANG AND MICEBONE

This subsection provides more results in addition to the ones presented in Subsection 5.2. In particular, Fig. 5 show the coverage - accuracy curves for NIH-AO, Chaoyang with 3 human experts and MiceBone.

### G.2 COMPARISON FOR DIFFERENT MISSING RATES

This subsection provides further results of our proposed method evaluated on Cifar-100 and Chaoyang at different missing rates.

**FixMatch L2D** Although FixMatch often surpasses MixMatch in several computer vision benchmarks, in the L2D setting shown in Fig. 3 and Fig. 5, FixMatch is on par with MixMatch and inferior in some settings. This is due to the strong data augmentation applied in real-world medical image datasets. The strong data augmentation in FixMatch is designed for natural images, and hence, when applied to medical images, it creates undesired artifacts or transformations, creating more difficulties for the learning algorithm to utilise unlabelled data (Zenk et al., 2022). In practice, the shearing

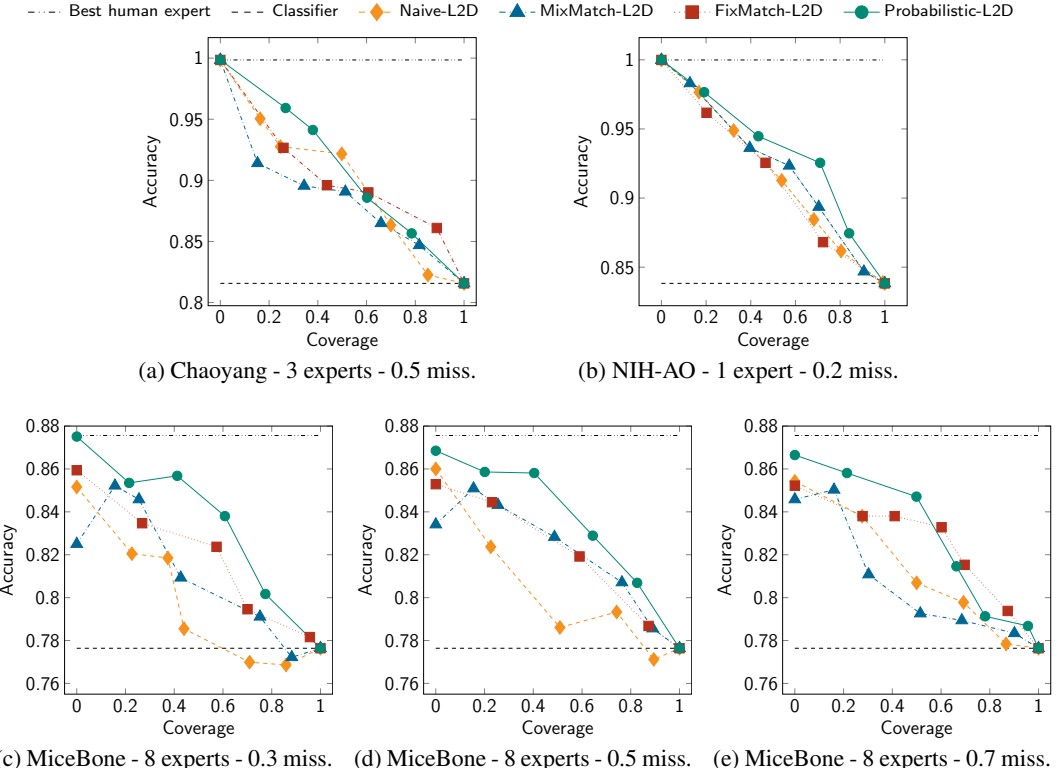

Figure 5: Additional coverage - accuracy curve results on: Chaoyang, NIH-AO and MiceBone.

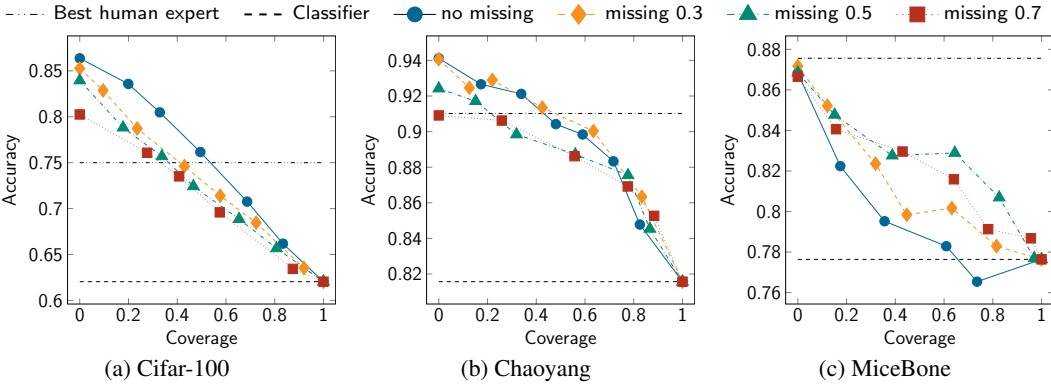

Figure 6: The coverage - accuracy curves of our method on different missing rates, including the setting with complete expert's annotation.

transformation in FixMatch is often omitted when applying on medical image datasets (Xu et al., 2022). In the paper, we use the vanilla FixMatch as a baseline in Section 5 to keep the consistency with previous studies (Hemmer et al., 2023).

# H    RUNNING TIME

We also report the running time to train the naive, semi-supervised baselines and our method in Table 3. These running times are reported for a whole training using the same number of epochs (1,000 for Cifar-100 and 300 for others). For the semi-supervised learning baseline, the running time includes the semi-supervised training for all experts (the first number) and the standard L2D (the second number).

Table 3: Running time of different L2D approaches (in GPU-hour); for semi-supervised learning baselines, the reported one includes the semi-supervised training for **all experts** (the first number) and the standard L2D training (the second number).

| Dataset | № experts | Running time (GPU-hour) | | | |
|---------|-----------|-------|----------|----------|------------------|
| | | **Naive** | **MixMatch** | **FixMatch** | **Probabilistic L2D** |
| Cifar-100 | 2 | 5.1 | 9.8 + 5.1 | 10.9 + 5.1 | 5.0 |
| Chaoyang | 2 | 1.8 | 5.4 + 1.8 | 5.8 + 1.8 | 3.8 |
| MiceBone | 8 | 4.4 | 22.4 + 4.4 | 25.3 + 4.4 | 12.8 |

# I  ABLATION STUDY ON THE MINI-BATCH SIZE FOR THE CONSTRAINED E STEP

We conduct another ablation study to understand the influence of mini-batch sizes to the constrained optimisation of the workload distribution presented in Section 4. Recall that the workload distribution is a global constraint, so larger batch sizes may lead to more accurate learning. We evaluate mini-batches of sizes 20, 50 and 100 at different missing annotation rates on Chaoyang dataset and plot the results in Figs. 7a and 7b. The area under curve of the evaluations presented in Fig. 7 agree with the intuition that the larger the batch size, the higher the performance.

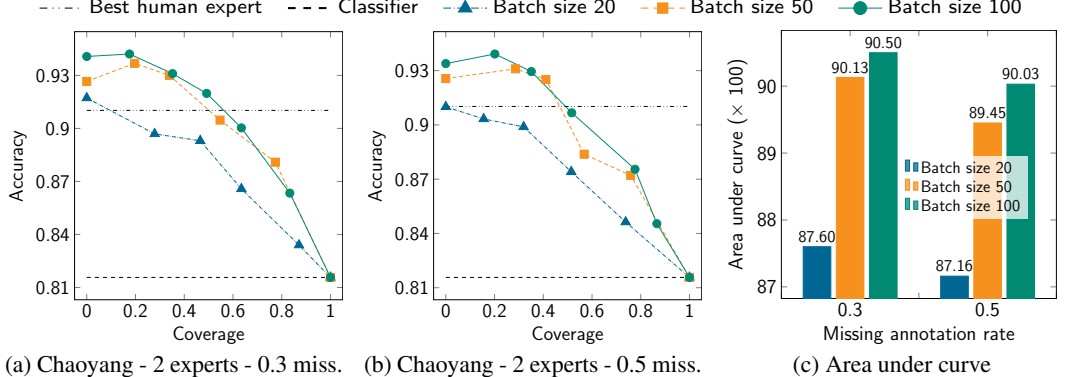

(a) Chaoyang - 2 experts - 0.3 miss.   (b) Chaoyang - 2 experts - 0.5 miss.   (c) Area under curve

Figure 7: Coverage - accuracy curves of probabilistic L2D trained on different mini-batch sizes and their corresponding area under curve.

# J  ONLINE EM

Conventionally, the EM algorithm requires to load and process the whole dataset at each iteration. This is, however, impractical, especially for large datasets that cannot fit into memory. Given this limitation, we employ an online version of the EM algorithm (Cappé & Moulines, 2009), which could be performed in a mini-batch setting. The online EM algorithm maintains an exponential moving average as an approximation of the full average $Q$ defined in Eq. (3). In particular, the stochastic EM algorithm can be presented as follows:

**E-step**  calculates the complete-data log-likelihood as the one in the conventional EM algorithm.

**M-step**  optimises the exponential weighted average w.r.t. the parameters of interest:

$$\gamma^{(k+1)}, \theta^{(k+1)} \leftarrow \underset{\theta, \gamma}{\mathrm{argmax}}(1 - \tau_k)Q\left(\theta, \gamma, \gamma^{(k-1)}, \theta^{(k-1)}\right) + \tau_k Q\left(\theta, \gamma, \theta^{(k)}, \gamma^{(k)}\right), \quad (33)$$

where $\tau_k$ is the step size satisfying the decaying conditions: $\sum_{k=1}^{+\infty} \tau_k = +\infty$ and $\sum_{k=1}^{+\infty} \tau_k < +\infty$.

Such a moving average update in the M step is similar to the momentum used in stochastic gradient descent. To this end, our implementation uses the SGD with a momentum of 0.9 as a solution that integrates the online EM into our method.

