# OpenReview forum: "Probabilistic Learning to Defer: Handling Missing Expert Annotations and Controlling Workload Distribution"
_ICLR.cc/2025/Conference — ICLR 2025 Oral_

### Official Review · Reviewer_edcU · 2024-10-22

**Soundness:** 3
**Presentation:** 3
**Contribution:** 3
**Rating:** 8
**Confidence:** 2

**Summary:**

The paper deals with human - AI cooperation. It presents a modification of the learning to defer technique which can handle incomplete expert annotations and balance the workload of the experts.

**Strengths:**

* It seems to me that the topic has been addressed very comprehensively
* The comparisons include all the mentioned relevant predecessor methods

**Weaknesses:**

I can't see any significant weaknesses. However, this may also be because the topic is new to me.

Further comments:

* In the case of “In contrast, machine learning or AI models excel at processing large amounts of information but may be prone to biases (Meehl, 1954)”, the reference chosen cannot be used as evidence for the statement because “machine learning or AI models ... processing large amounts of information” were not available until long after 1954.
* I find statement “Ideally, a perfect balanced workload among experts and the AI model can be expressed as follows” a little strange. After all, you will only strive for an equal distribution if all experts are equally competent.
* I wonder about “slightly-similar”, how can something be slightly similar?
* I find it a bit irritating that there is no section called “Conclusion”.
* “50 %” -> “50%”

**Questions:**

* Would it make sense to repeat the learning experiments several times in order to be able to estimate the uncertainty of the results?
* Does it make sense to give the results in Table 1 with four valid digits? Are the results really that accurate?

---

> ### Author Response · Authors · 2024-11-20
> **Repeating experiments to estimate the uncertainty**
>
> As mentioned in our paper, each L2D method has to be run multiple times to construct its coverage - accuracy curve to analyse the trade-off between coverage and prediction accuracy (or, reliability versus operating cost mentioned in the introduction of our paper). We can also run multiple times at each targeted coverage to obtain a finer estimation of uncertainty. In that case, each point on a coverage - accuracy curve will be represented by two error bars: one along the x-axis denoting the variation of coverage and the other along y-axis denoting the variation of prediction accuracy. Although that provides a fine-grain uncertainty estimation, it is costly and time-consuming. Furthermore, that uncertainty complicates the calculation of area-under-curve. That is why we decide to follow a simple solution, which is to construct the coverage - accuracy curves only.

---

> ### Author Response · Authors · 2024-11-20
> **Number of significant digits when reporting results**
>
> To ensure consistency with the number of testing samples across all datasets, we employ a fixed number of significant digits in our reported results. As the number of testing samples varies from 1,000 (e.g., NIH-AO, Chaoyang, MiceBone) to 10,000 (e.g., Cifar-100), a precision of three or four digits is necessary. To maintain uniformity in Table 1, we have standardised the reporting to four significant digits.

---

> ### Author Response · Authors · 2024-11-20
> **Minor revision on references, typos and explanation**
>
> We thank the reviewer for all the minor issues in the initial review. We have accummulated those issues and updated in our revision. In particular:
> - For the reference: We apologise for overlooking this matter. We removed the reference in our revision.
> - For the statement of workload constraint: We revise the sentence as follows: *"Ideally, in many applications where human experts are equally-competent, a perfect balanced workload among experts can be expressed as follows:"* in our revision.
> - Similar experiment setting: What we meant is to simulate asymmetric label noise similar to (Hemmer et al., 2023). However, instead of using *strength* and a pre-trained EfficientNet-B1, we  simulate label noise based on the common setting of asymmetric label noise in noisy label learning. To increase clarity, we remove the word *slightly* in our revision.
> - Missing conclusion: We created a new section for discussion of our limitations and dedicating the last section for concluding. Please see our revision for this change.
> - Typos: We corrected all the typos detected after our initial submission in our revision.

---

> > ### Comment · Reviewer_edcU · 2024-11-24
> > **Current status**
> >
> > I have read the authors' answers and the reviews of the other reviewers and see the paper positively. Unfortunately, none of the reviewers has high confidence.

---

> > > ### Author Response · Authors · 2024-11-26
> > > **Thank you for responding**
> > >
> > > We thank the reviewer for engaging into the discussion and considering our paper positively. We wonder if there are any concerns from the reviewer about our response to the reviewer's initial feedback as well as the latest revision of our paper.

---

### Official Review · Reviewer_1P43 · 2024-10-31

**Soundness:** 3
**Presentation:** 3
**Contribution:** 3
**Rating:** 8
**Confidence:** 3

**Summary:**

This paper introduces a probabilistic framework for “Learning to Defer” (L2D) that enables an AI system to either make decisions independently or defer them to human experts based on confidence levels. The approach addresses the limitations of existing L2D methods by handling cases where not all experts can annotate every data sample. This approach is designed to reduce the annotation burden. The approach is based on the Expectation-Maximization (EM) algorithm to optimize workload distribution between AI and human experts. The proposed method shows promising performance across synthetic and real-world datasets.

**Strengths:**

- The paper addresses an interesting issue in L2D and proposes a sound solution based on a probabilistic approach.
- The workload management is particularly promising in many areas where AI is supporting expert decision such as in medicine.
- This is also relevant in addressing ethical and practical constraints, and possibly even regulations and laws.
- The ablation study offers an insight on the mechanism that lead to prioritise highest performing humans with the imbalanced approach, with possible overfitting.
-  It is interesting that the study allows for the conclusion that in practice it may be desirable to distribute workload evenly across all human experts.

**Weaknesses:**

1. Overall, the approach has some limitations, which I acknowledge are also partially discussed. However, it's unclear how well the system can scale given that each expert requires a probabilistic model. It's unclear to me how well the clustering of expert would work and what are the risks associated with that.

2. I would be interested in reading more about the trade-off between the case for fewer deferring cases or deferring cases with the highest uncertainty, which is not much discussed. Clearly, there will be cases, e.g., healthcare, where deferring on uncertain cases would be quite important.

3.  How could the model be adapted to take into consideration fast and slow changing expertise performance? The model assume static performance, however, experts could have fast performance changes, e.g. due to fatigue, or slow performance changes, e.g. due to learning through a period of time. It would be nice to understand how the model could accommodate for such dynamic scenarios.

**Questions:**

Could the authors answer points/questions 2 and 3 in the list above "weaknesses"?

---

> ### Author Response · Authors · 2024-11-20
> **Discussion on the trade-off between low and high coverage**
>
> We thank the reviewer for the constructive suggestion. As we mentioned in the paper, L2D aims to maximise reliability, while keeping the cost within the acceptable limits. The choice of the ideal operating point that balances reliability and costs  typically depends on specific priorities. In high-stakes applications (e.g., healthcare), deferring uncertain cases offers significant advantages, such as improving decision accuracy, enhancing trust by acknowledging AI limitations, and leveraging human oversight to ensure accountability in high-stakes scenarios. Deferred cases can also provide valuable learning opportunities to improve the AI model. However, deferring cases can be costly due to increased annotation budgets, potential expert burnout, and possible delays in decision-making.
>
> On the other hand, minimising deferrals maximises the efficiency of the AI system, allowing for lower costs and quicker decision-making. This approach also reduces reliance on human experts and demonstrates the AI's ability to handle easier cases independently. However, not deferring uncertain cases poses risks, such as potential misdiagnoses and diminished oversight. These risks can have severe consequences for patient safety and undermine trust in the system, especially when high-stakes decisions are involved.
>
> An optimal approach involves adaptive strategies that balance costs, risks, and accuracy. This strategic management allows medical AI systems to maintain high standards of care while remaining cost-effective.
>
> We also included this discussion in section 7 of our revision.

---

> ### Author Response · Authors · 2024-11-20
> **Performance change of human experts**
>
> We thank the reviewer for the suggestion. We agree with the reviewer that the current setting of L2D assumes a static performance of each human expert, although this is not true in practice.
>
> To accommodate dynamic expert performance, the model could be extended to incorporate temporal dynamics and performance variability by integrating sequential modelling techniques or adaptive mechanisms. For fast-changing performance, such as fatigue, the model could include real-time performance tracking using metrics such as dynamic accuracy, allowing the adjustment of the deferral strategy based on current conditions. Slow-changing performance, like expertise improvement through learning, can be modelled using techniques such as Bayesian updating to adapt expert-specific parameters over time. Temporal modelling (e.g., recurrent neural networks) can also capture patterns in performance changes, enabling the system to make adjustments. By integrating such adaptive mechanisms, the model would better reflect the short-range and long-range evolving nature of human expertise, improving its decision-making capabilities in dynamic scenarios.
>
> In our revision, we included the discussion on the performance change of human experts in Section 7.

---

> ### Author Response · Authors · 2024-11-20
> **Clustering human experts and risks associated**
>
> We thank the reviewer for the comment. We are actively looking at clustering human experts as a solution to address the scalability of our proposed method. As mentioned in the paper, by clustering $M$ human experts into $K$ groups, where $M \gg K$, we can reduce from a large number of $M$ models to a much small number of $K$ models to represent each group of human experts.
>
> We acknowledge that there are some risks associated with that clustering approach. For example:
>  - Clusterability: which indicates whether data representing the pattern of labelling can be separated and grouped into clusters,
>  - Optimality: the clustering may over-simplify user labelling behaviour into coarse clusters that do not represent well the users.
>
> These risks can be mitigated by the collection of data that make the labelling patterns of human experts distinguishable. Alternatively, we can try the following options: i) perform a study on the number of clusters, and ii) put in place a rigorous cluster validation with quantitative metrics (e.g., silhouette score) and qualitative checks (e.g., expert domain validation).

---

> > ### Comment · Reviewer_1P43 · 2024-11-27
> >
> > I thank the authors for providing valid answers to my comments and commit to improve the paper accordingly. In light of those, I maintain my positive assessment of the paper.

---

### Official Review · Reviewer_6swQ · 2024-11-03

**Soundness:** 3
**Presentation:** 3
**Contribution:** 3
**Rating:** 8
**Confidence:** 2

**Summary:**

The paper studies the problem of learning to defer for a realistic setting where expert annotations can be missing and where workload balancing among experts is crucial. The authors propose a probabilistic modeling approach where EM is used to address the missing annotations, In particular, a constrained optimization during the E step regulates workload balancing among human experts and the AI classifier. The proposed method is evaluated on synthetic and real-world datasets and is shown to perform on par or better than the considered baselines.

**Strengths:**

The paper is well-written and easy to follow. The proposed probabilistic modeling techniques and the use of EM in this setting seem novel and an interesting contribution. Experimental results show the performance gain of the method compared to the baselines.

**Weaknesses:**

A key weakness is highlighted by the authors in the paper: Bad dependency on the number of human experts. Although, they discuss potential remedies, e.g., clustering. However, this probably wouldn't work for a setting with diverse human experts (where the number of clusters is large). Are there other dimensionality reduction approaches (e.g., hierarchical clustering) that one could consider for this setting and how would they affect computational cost?

**Questions:**

I am unsure about the computational cost of the method compared to the considered baseline. Could the authors elaborate more on this? Ideally, by comparing and reporting the runtimes of the proposed method and the baselines across the different datasets.

---

> ### Author Response · Authors · 2024-11-20
> **Potential remedies to address the scalability**
>
> We agree with the reviewer that in case of diverse human experts, the number of clusters may be large, which requires the application of further dimensionality reduction (e.g., hierarchical clustering as suggested by the review). Alternatively, we can use a *conditional* model $h(\mathbf{x}, \zeta)$, where $\zeta$ is the embedding of a human expert, to model each human expert. The embedding of each human expert $\zeta$ can be obtained by extending the sample-wise representation learning to set-wise representation learning (e.g., using contrastive learning coupled with "deep set" (NIPS 2017)). Such a conditional model allows the scalability of our method to a large number of human experts.
>
> We included the discussion on the diverse number of human experts and conditional modelling in our revision and highlighted it in blue.

---

> ### Author Response · Authors · 2024-11-20
> **Computational cost**
>
> We reported the running time of each method in Table 3 of the Appendices of our original submission. To make it clearer to reader, we revise our paper and explicitly refer to Table 3 as the computational time when discussing our limitations (please refer to line 495 in section 7 of our revision).

---

> > ### Comment · Reviewer_6swQ · 2024-11-20
> > **Response to rebuttal**
> >
> > Thanks for addressing my concerns. I have read the author's response to my concerns and increased my score to 8. I am not a specialist in this field therefore I have kept the confidence of my review low.

---

> > > ### Author Response · Authors · 2024-11-26
> > > **Thank you**
> > >
> > > We thank the reviewer for engaging into our discussion and updating their rating of our paper.

---

### Official Review · Reviewer_CxS1 · 2024-11-03

**Soundness:** 3
**Presentation:** 4
**Contribution:** 3
**Rating:** 8
**Confidence:** 3

**Summary:**

This paper extends the concept of *learning to defer* (L2D) to scenarios with missing expert annotations and balanced expert workloads. The authors propose a formulation that relies on a clever application of the expectation-maximization algorithm, which naturally handles missing data. Additionally, they introduce a constraint within the expectation stage of the algorithm to manage expert workloads. The proposed L2D is tested on both synthetic and real-world datasets, resulting in a higher area under the coverage-accuracy curve compared to the evaluated baselines.

**Strengths:**

+ The paper addresses a research question relevant to real-world applications by providing a solution for settings where expert annotations are incomplete.
+ The results show that reducing the workload of highly accurate (and typically overloaded) human experts only slightly decreases overall accuracy and can lead to higher accuracy in scenarios with inconsistent expert performance between the training and test sets.
+ The proposed controllable workload formulation simplifies the evaluation of accuracy-coverage ratios compared to existing methods, which often require assumptions or post-hoc adjustments to balance learnable models and human experts.

**Weaknesses:**

+ As acknowledged by the authors, the proposed formulation does not scale well with the number of human (or learnable) experts. While grouping experts into clusters is suggested as potential future research direction, this introduces the number of clusters as a hyperparameter, necessitating additional tuning and potentially hindering scalability.
+ Although the paper is concise and generally well-written, the notation is ambiguous in some places (see Q1 and Q2), and the discussion of the results is very brief and could benefit from additional explanations (see Q3 and Q4).
+ (Minor comment) I recommend the authors release the source code to reproduce results. While not mandatory, providing the code would help readers understand how to implement the algorithm proposed on page 14, especially the implementation steps required to solve the optimization equation formulated in Eq. 4 on page 4.

**Questions:**

+ Q1. I found the data training process and the L2D objective on page 3 difficult to understand, possibly due to ambiguous notation. Does $\mathbf{y}$ represent the ground truth label or output prediction? If $\mathbf{y}$ refers to the ground truth, why does it depend on the expert annotations $\mathbf{t}$? Should not $\mathbf{y}$ be dependent on $\mathbf{x}$ but independent of expert selection and annotation? The notation in Eq. 1, where $\mathbf{t}_i$ becomes $\mathbf{y}_i$, is particularly confusing. Revising the notation could improve clarity and help readers understand the logic.
+ Q2. Regarding Eq. 1, if $\mathbf{t}_i$ are deterministic expert annotations from a look-up table and $\mathbf{y}_i$ represents the ground truth label, is it still valid to compute the log-likelihood with *hard labels*?
+ Q3. Could you provide a more detailed explanation of why the accuracy of probabilistic L2D surpasses that of the best human expert (even with a 70% missing rate and especially for low coverages)? This was not explained in the discussion.
+ Q4. Why does the area under the coverage-accuracy curve increase with higher missing rates on the MiceBone dataset? Why does having fewer annotations lead to better performance? It is mentioned that this is due to inconsistent human expert behavior between the training and test sets. How do missing annotations help in this case?
+ Q5. The discussion on balanced workload and inconsistent human performance between training and test sets could benefit from further elaboration (Page 8, Line 423). Would assigning less weight to a particular expert act as a form of regularization?
+ Q6. (Minor suggestion) It would be helpful to present the results in Figure 3 with equal Y-axis scales for subplots corresponding to the same dataset.

---

> ### Author Response · Authors · 2024-11-20
> **Ambiguous notation and dependency between ground truth y and input x**
>
> As mentioned at line 107, $\mathbf{y}$ represents ground truth label, and $\mathbf{t}$ represents annotations of human experts. In our modelling, $\mathbf{y}$ is derived from expert annotations $\mathbf{t}$ and selection $\mathbf{z}$. While $\mathbf{x}$ provides the input context, the model assumes that all relevant information from $\mathbf{x}$ is captured via $\mathbf{t}$ and $\mathbf{z}$, making $\mathbf{y}$ conditionally independent of $\mathbf{x}$. This design prioritises modelling expert-driven uncertainty and subjectivity, aligning with real-world annotation practices.
>
> The confusion at line 150 between $\mathbf{y}$ and $\mathbf{t}$ is due to the need to train the gating model and the classifier at the same time. To simplify, an equivalent procedure is to remove the second term from Eq. (1) (from the submitted paper) by training the classifier using ground truth labels. In that case, the trained classifier is considered as one expert. In the subsequent step, Eq. (1), from the submitted paper, consists of only the first term to train the gating model, and $\mathbf{t}$ is simply the annotations made by experts (either human or the classifier) without being modified. For clarification, we add an explanation in our revision and highlight in blue.

---

> ### Author Response · Authors · 2024-11-20
> **Compute log-likelihood when annotations are hard labels**
>
> In theory, the log-likelihood of the ground truth label $\mathbf{y}$ evaluated on the categorical distribution parameterised by annotation $\mathbf{t}$ is still valid even though $\mathbf{t}$ is a one-hot vector (the log-likelihood is either 0 or negative infinity at the limit). However, that may lead to numerical instability in practice. Hence, in our implementation, we smooth the annotation $\mathbf{t}$ to avoid such numerical issue.

---

> ### Author Response · Authors · 2024-11-20
> **The accuracy of probabilistic L2D surpasses the best human expert**
>
> The high performance that surpasses the best human expert at low coverage is attributed by the learnt deferral mechanism of the gating model, which selects the correct human expert to make decision. Indeed, this is one of the motivations to study the human - machine cooperation mentioned in the introduction of our paper.
>
> To make it clearer, we note that the human experts considered in each experiment provide complementary label correctness, compared to each other. For example, in Cifar-100, each expert makes mistakes on non-overlapping super-classes, and hence:
>         \begin{equation}
>             \begin{aligned}[b]
>                 \text{Even though }\Pr(\mathbf{t}^{(1)} = \mathbf{y}) \approx \Pr(\mathbf{t}^{(2)} = \mathbf{y}) \approx 0.75,
>                 \text{we can have }\Pr(\mathbf{t}^{(1)} = \mathbf{y} \quad \operatorname{OR} \quad \mathbf{t}^{(2)} = \mathbf{y}) = 1,
>             \end{aligned}
>         \end{equation}
>         Ideally, a perfect gating model will defer each query to the correct human expert, reaching 100\% prediction accuracy (when their label correctness is complementary). That explains why in Cifar-100 and Chaoyang, the L2D system performs even better than the best human expert at low coverage. Note that, in Figure 3, the results are lower than 100\% is because 1) *imperfect gating model*: the gating model is not perfect and cannot always select the correct human experts, and 2) *missing annotations* introduces uncertainty, resulting in difficulty for the gating model to defer accurately.

---

> ### Author Response · Authors · 2024-11-20
> **Area under the coverage - accuracy curve increases with higher missing rate on MiceBone**
>
> We agree with the reviewer that the area under the coverage - accuracy curve increases with higher missing rates on MiceBone across all methods. This is due to the missing data coupled with the inconsistent performance of human experts in training and testing.
>
> For example, let's consider the naive baseline between two cases: 1) no missing label, and 2) 70\% missing labels. Without any regularisation, the naive L2D will often select the *best* human expert observed in the training set. According to the updated Table 2 presented below (and also  updated in our revision in Appendix F), which presents the performance of human experts in training and testing, the naive L2D will most likely select expert 534 in the case of no missing label, and expert 290 in the case of 70\% missing label due to their highest performance in each setting (see the updated Table 2 below). This coupled with the inconsistent performance in the testing set (see the column *Test* in the same table) leads to the observation: lower performance at lower missing rate during testing. For example, 84.45\% prediction accuracy when there is no missing annotation versus 85.29\% at 70\% missing rate. In our revision, we add a discussion in the section of experiments to explain this observation on MiceBone.
>
> | Expert id  | Train - 0% missing | Train - 30% missing | Train - 50% missing | Train - 70% missing | Test |
> | --- | --- | --- | --- | --- | --- |
> | 047 | 86.34 | 64.45 | 49.15 | 34.68 | 85.74 |
> | 290 | 85.82 | 65.63 | 49.82 | **36.81** | 85.29 |
> | 533 | 86.29 | 64.37 | 49.61 | 35.30 | 84.96 |
> | 534 | **86.59** | 64.47 | 50.11 | 35.00 | 84.45 |
> | 580 | 80.76 | 60.40 | 47.90 | 33.79 | 79.07 |
> | 581 | 85.83 | 64.12 | 50.43 | 36.37 | 84.32 |
> | 966 | 86.54 | 64.47 | 49.41 | 35.62 | 87.56 |
> | 745 | 85.01 | 65.61 | 50.71 | 35.81 | 84.38 |

---

> ### Author Response · Authors · 2024-11-20
> **Assigning less weight to a particular expert acts as a form of regularization**
>
> In fact, imposing workload constraint is a form of injecting prior knowledge into the estimation of posterior of latent variables, resulting in a form of regularisation. Different prior knowledge to define workload distribution will result in different regularisation effects. As for the question raised by the reviewer, assigning smaller weight to a particular expert is equivalent to integrate a prior into training, and hence, it is equivalent to a regularisation.

---

> ### Author Response · Authors · 2024-11-20
> **Scaling y-axes of subplots and code release**
>
> We thank the reviewer for the constructive suggestion to improve the paper. We revised the paper by scaling the y-axes of subplots in Figures 3 and 5 to increase the clarity in the qualitative analysis. Please refer to our latest revision uploaded to Open Review for this change.
>
> We also provide a link to an anonymous repository of our implementation in our revision. Please refer to that repository for the details of our implementation.

---

> > ### Comment · Reviewer_CxS1 · 2024-12-02
> > **Updated Review**
> >
> > The authors have addressed all my comments by revising the notation, including additional discussions, and providing the source code for reproducibility. Based on the revised paper and the responses, I have updated the score.
> > Good luck with your work!

---

> > > ### Author Response · Authors · 2024-12-04
> > > **Thank you for helping to improve the clarity of the paper**
> > >
> > > We thank the reviewer CxS1 for the constructive feedback to help improving the clarity of our paper. We also appreciate for considering our paper positively.

---

### Author Response · Authors · 2024-11-20
**Summary of changes made in the revision**

We thank all reviewers for their insightful and constructive feedback. We have carefully considered the comments from each reviewer and have incorporated these suggestions into a revised version of our paper, which has been uploaded to Open Review. All modifications are highlighted in blue for easy reference. Specifically, we have:
- enhanced the clarity of the paper by providing more detailed explanations of the modeling techniques and experimental results,
- separated the discussion from conclusion to address the limitations of the proposed method and the current setting of learning-to-defer, and
- corrected some typos and visualisation (scaling y-axes of subfigures).

We have also addressed each reviewer's concerns individually, which can be found directly below their respective reviews.

---

### Author Response · Authors · 2024-12-04
**Thank you for constructive feedback and positive review**

We thank all the reviewers for their constructive feedback to improve our paper. In particular:
 - we appreciate the initial review of reviewer 1P43 for praising our paper and suggesting insightful discussion on many practical aspects of the setting considered in the paper,
 - we thank the reviewer 6swQ for suggesting and discussing further potential solutions to tackle the limtiation of the proposed method,
 - we thank the reviewer CxS1 for helping to improve the clarity of the methodology of the proposed method, as well as suggesting to add  intuitive explanation on our empricial evaluation, and
 - we also thank the reviewer edcU for helping to make our writing precise and easier to understand.

---

### Meta-Review · Area_Chair_GuFp · 2024-12-14

**Metareview:**

This paper addresses the problem of learning to defer in scenarios with incomplete expert annotations and the need for balanced expert workloads. The reviewers generally agree that the paper is well-written, addresses an interesting and relevant problem, and presents a novel and sound solution. The workload management aspect is seen as particularly promising for applications where AI supports expert decision-making, such as in healthcare. The experimental results show the performance gains of the proposed method compared to baselines. Meanwhile, some weaknesses and limitations are noted. The method's scalability with respect to the number of human experts is a concern, and while the authors discuss potential remedies like clustering, further investigation is needed to assess their effectiveness, especially in settings with diverse experts. The trade-off between deferring fewer cases versus deferring the most uncertain cases could be discussed in more detail. Additionally, the current model assumes static expert performance, and adapting it to handle fast and slow-changing expertise performance would be valuable. Overall, the reviewers considered it a good contribution to the field.

**Additional Comments On Reviewer Discussion:**

The authors have addressed the reviewers' comments and questions in their rebuttal, providing clarifications and proposing revisions to improve the paper. They have discussed the trade-off between low and high coverage, potential approaches to accommodate dynamic expert performance, and the risks associated with clustering experts.

---

### Decision · Program_Chairs · 2025-01-22

Accept (Oral)